



# CO₂, CH₄ and N₂O fluxes along an altitudinal gradient in the northern Ecuadorean Andes: N₂O consumption at higher altitudes

Paula Alejandra Lamprea Pineda[1], Marijn Bauters[1, 2], Hans Verbeeck[2], Selene Baez[3], Matti Barthel[4] and Pascal Boeckx[1]

[1]Isotope Bioscience Laboratory – ISOFYS, Department of Green Chemistry and Technology, Ghent University, Gent, 9000 Belgium

[2]Computational and Applied Vegetation Ecology – CAVElab, Department of Environment, Ghent University, Gent, 9000 Belgium

[3]Departamento de Biología, Escuela Politécnica Nacional del Ecuador, Ladrón de Guevera E11-253 y Andalucía, Quito, Ecuador

[4]Department of Environmental Systems Science, ETH Zurich, Zurich, 8092, Switzerland

*Correspondence to*: Paula Lamprea (paulaalejandra.lampreapineda@ugent.be)

Co-authors: marijn.bauters@UGent.be, hans.verbeeck@UGent.be, selene.baez@epn.edu.ec, matti.barthel@usys.ethz.ch, pascal.boeckx@UGent.be

Keywords: Tropical Forests - Soil CO₂, CH₄ and N₂O fluxes - Altitudinal Gradients - Source-partitioning N₂O - Stable Isotopes.

## Abstract

Tropical forest soils are an important contributor to the global greenhouse (GHG) budget and understanding this ecosystem function is of vital importance for future global change and climate research. In this study, we quantified soil fluxes of carbon dioxide (CO₂), methane (CH₄) and nitrous oxide (N₂O) of four tropical forest sites located along an altitudinal gradient from 400 to 3010 m a.s.l. on the western flanks of the Andes in northern Ecuador. We assessed the physicochemical soil properties influencing these fluxes during the dry season, as well as the bulk isotopic signature of N₂O. The CO₂ fluxes ranged between $55.3 \pm 12.1$ and $137.6 \pm 32.8$ mg C m$^{-2}$ h$^{-1}$, with the highest and lowest emissions at the highest strata, at 3010 and 2200 m a.s.l., respectively. CH₄ fluxes at all sites exhibited a net consumption of atmospheric CH₄ and ranged between $-74.4 \pm 25.0$ µg C m$^{-2}$ h$^{-1}$ at 2200 m a.s.l. to $-46.7 \pm 14.7$ µg C m$^{-2}$ h$^{-1}$ at 3010 m a.s.l. Net fluxes of N₂O ranged between $-5.1 \pm 1.9$ and $13.2 \pm 31.3$ µg N m$^{-2}$ h$^{-1}$, with a marked net sink at 2200 and 3010 m a.s.l., whereas a net source at 400 m. pH$_{water}$ and nitrate (NO₃⁻) content at 5 cm depth were able to explain 83% of the observed temporal (daily measurements) and spatial (four forest sites) variability of the CO₂ fluxes; indicating that an increase in pH$_{water}$ and NO₃⁻ contents lead to an increase in CO₂ emissions. For CH₄ fluxes, it was not possible to obtain a statistically significant model to identify the physicochemical soil drivers responsible for the CH₄ consumption. For N₂O, bulk density and pH$_{water}$ at 5 cm depth were negatively correlated to the N₂O fluxes, but able to explain only 36% of the temporal and spatial variability. In addition, the bulk isotope N₂O data confirmed that N₂O reduction was at the basis of the observed net soil sink at higher altitudes. Finally, the soil GHG budget showed that all studied soils were net sources of GHG's. CO₂ emissions represented the largest component of the total soil GHG budget, CH₄ consumption was quite consistent along the elevation gradient, whereas N₂O was highly variable, and the transition from sources to net





sinks at higher altitudes represented the biggest change in the net GHG balance. Overall, for non-$CO_2$ GHGs, we noticed a

transition from a net source to a net GHG sink along altitude.

## 1 Introduction

In 2014, the Intergovernmental Panel on Climate Change (IPCC) released its Fifth Assessment Report (AR5) indicating that the atmospheric concentrations of the three major biogeochemical greenhouse gases (GHGs) (carbon dioxide - $CO_2$, methane - $CH_4$ and nitrous oxide - $N_2O$) have reached unprecedented levels. As such, current atmospheric concentrations indicate that

in 2018, $CO_2$ (407.8±0.1 ppm), $CH_4$ (1869±2 ppb) and $N_2O$ (331.1±0.1 ppb) concentrations were 147%, 256% and 123%, higher compared to pre-industrial levels (before 1750) (WMO, 2019). A large proportion of $CO_2$ emissions are sourced by human activities, particularly fossil fuel burning and land use change. Similarly, the increase in $CH_4$ emissions is mainly due to fossil fuel and agriculture (60%), as well as for $N_2O$, of which food production is the largest contributor to the $N_2O$ increase (Syakila and Kroeze, 2011).


Soils in terrestrial ecosystems play a vital role in the global GHG budget. Tropical forest soils, in particular, represent a net sink of carbon (C) (Pan et al. 2011), but they coincidently are the largest natural source of $N_2O$, with an estimated contribution of 14-23% to the annual, global $N_2O$ budget (Werner et al., 2007). In general, soil $CO_2$, $CH_4$ and $N_2O$ production or consumption depends on microbiological processes driven by a wide range of abiotic and biotic characteristics. The

combination of these processes ultimately determines if a soil is a net source or sink of GHGs. Under aerobic conditions, $CO_2$ is emitted to the atmosphere by autotrophic and heterotrophic respiration of vegetation and fauna (Dalal and Allen, 2008), while $CH_4$ is consumed by methanotrophic bacteria (Jang et al., 2006); although forest soils prone to inundation (anaerobic) emit $CH_4$ by methanogenic microorganisms (*Archaea* domain). On the other hand, $N_2O$ is emitted through denitrification or a number of alternative pathways (e.g. nitrification, nitrifier-denitrification, chemodenitrification, etc.; (Butterbach-Bahl et al.,

2013; van Cleemput, 1998; Clough et al., 2017)). Overall, tropical forests emit on average 12.1 t $CO_2$-C ha$^{-1}$y$^{-1}$ (heterotrophic and autotrophic respiration), slightly smaller than the Net Primary Productivity (NPP) (12.5 t $CO_2$-C ha$^{-1}$y$^{-1}$) i.e. the net C sink of tropical forests is ~ 0.4 t $CO_2$-C ha$^{-1}$y$^{-1}$. In aerobic conditions, $CH_4$ fluxes vary from -1 to -40 kg $CH_4$ ha$^{-1}$y$^{-1}$, with an average consumption of -4 kg $CH_4$ ha$^{-1}$y$^{-1}$, while the mean rate of $N_2O$ emissions from tropical forest soils is 4.76±0.81 kg $N_2O$ ha$^{-1}$y$^{-1}$ (Dalal and Allen, 2008), i.e. 2-3 times higher than the mean $N_2O$ emissions from temperate forest soils (1.57±0.56 kg $N_2O$

ha$^{-1}$y$^{-1}$; Chapui-Lardy et al., 2007; Van Groenigen et al., 2015). Despite de lower concentrations compared to $CO_2$ and $CH_4$, $N_2O$ emissions have received increasing attention over the last decades because 1) the global warming potential of $N_2O$ is almost 300 times that of $CO_2$, (IPCC, 2014b; Myhre et al., 2013), 2) $N_2O$ emissions contribute to the depletion of the stratospheric ozone layer (Portmann et al., 2012), and 3) they remain in the atmosphere for approximately 130 years (Myhre et al., 2013).




The understanding of the mechanisms and processes underlying GHG flux variability has greatly improved during the last decades (Butterbach-Bahl et al., 2013; Heil et al., 2016; Müller et al., 2015; Sousa Neto et al., 2011; Su et al., 2019; Teh et al., 2014). However, there is still (1) considerable uncertainty about the overall balances of many ecosystems (Castaldi et al., 2013; Heil et al., 2014; Kim et al., 2016; Pan et al., 2011; Purbopuspito et al., 2006), (2) a strong imbalance in field observations,

skewed to the Northern Hemisphere (Jones et al., 2016; Montzka et al., 2011), and (3) a bias towards quantification of emissions in lowland forests within the tropics (Müller et al., 2015; Purbopuspito et al., 2006). For instance, based on a compilation made of $CO_2$, $CH_4$ and $N_2O$ fluxes in South America (Table S.3), there are only two studies on emissions in upper montane forests, while they represent 11% of the world's tropical forests (Müller et al., 2015; Teh et al., 2014). To further improve our understanding of tropical forest ecosystems on the global GHG balance, environmental gradients (elevational,

latitudinal, etc.) can offer great opportunities to study the influence of abiotic factors on biogeochemical processes under field conditions (Bauters et al., 2017a; Jobbágy and Jackson, 2000; Kahmen et al., 2011; Laughlin and Abella, 2007), which complements the knowledge on short term responses from experimental approaches. In the case of elevational gradients, these responses are driven by abiotic variables that co-vary with elevation, which, amongst others, creates a distinctly strong climate gradient over a short spatial distance (Bubb et al., 2004; Killeen et al., 2007; Körner, 2007; Myers et al., 2000).


Here, we present a study of the soil-atmosphere exchange of $CO_2$, $CH_4$ and $N_2O$ along an altitudinal gradient in a Neotropical montane forest located on the western flanks of the Andes in northern Ecuador. We aimed (1) to determine the magnitude of the soil-atmosphere exchange of $CO_2$, $CH_4$ and $N_2O$ during the dry season, and (2) to assess the main climatic and soil parameters that control these fluxes. By working along this altitudinal gradient, we wanted to explore the potential effect of

temperature - and other factors that co-vary with altitude - on the GHG budget of the forest soils. We expected the $CO_2$ fluxes to decrease with altitude, with higher emissions at lower altitudes in view of more ideal conditions (i.e. temperature and soil moisture); $CH_4$ fluxes to represent net sinks that increase (i.e. negative fluxes) with altitude, and mainly explained by soil moisture; and a decrease in $N_2O$ fluxes along the altitudinal gradient, with soil moisture and nitrogen (N) content as the main explanatory variables.

## 90    2 Materials and methods

### 2.1 Study areas

The field work was carried out along an altitudinal gradient from lowland (400 m a.s.l.) to upper montane evergreen forests (3010 m a.s.l.; Table 1) (FAO, 2017; Ministerio del Ambiente, 2015). We selected areas of well-preserved natural forests located on the western flanks of the Andes in northern Ecuador; specifically, in the Sierra region of the provinces of Imbabura

and Pichincha. Four study sites (Fig. S1) were selected: Río Silanche at 400 m a.s.l (hereinafter: S_400), Milpe at 1100 m a.s.l. (hereinafter: M_1100), El Cedral at 2200 m a.s.l. (hereinafter: C_2200) and Peribuela at 3010 m a.s.l. (hereinafter: P_3010). Observations were made within one plot of about 20x20 m, established at each study site.



All sites experience two rainy seasons (March - April and October - November), with a mean annual precipitation (MAP) that
varies on average between 900 and 3600 mm, and an adiabatic lapse rate of approx. 5 °C per 1000 m of altitude (Table 1)
(Varela and Ron, 2018).

## 2.2 Sampling strategy

At all study areas, the sampling campaign took place from August 6[th] to September 28[th], 2018, corresponding to the end of the
dry season. One plot was selected for each site, and within each plot, five polyvinyl chloride (PVC) collars were installed to
allow *in-situ* measurements using a static flux chamber method. The collars were inserted at random locations within the plots
but guaranteeing at least 7 m distance between each one. The insertion of the collars was performed at least 12 h before the
first measurements by applying even pressure across all points to minimize effects caused by soil disturbance. The chambers
consisted of a PVC pipe hermetically sealed on top with a rubber-sealed lid. The chamber area was 0.0191 m$^2$ and the internal
volume ranged between 3.63 and 3.98 L. Each chamber was equipped with sampling ports mounted with three-way valves,
and a vent tube was installed to reduce pressure interferences.

Gas samples were collected mid-morning and measurement cycles on each site consisted of four consecutive gas measurements
once per day for one hour and during five contiguous days. The samples were taken mid-morning to avoid extreme
temperatures and we consider them as representative of a whole day (Collier et al., 2014; Luo and Zhou, 2006d). For these
measurements, the collars were left in place for the duration of each measurement cycle; thus, the analysis per stratum lasted
1 week (i.e. 1 month for all measurements in the 4 strata). However, in order to assess a both short-term and long-term variation
mainly related to weather conditions, the gas measurements were done first in August and consequently repeated in the next
month (September).

Adjacent to each chamber (~ 1 m), one pit was dug for soil sampling, and intact soil cores were collected using stainless steel
cylinders (diameter: 5.08 cm, height: 5.11 cm). The samples were taken at 5 and 20 cm depth once during the first month
(August) of measurements. Each soil core was immediately packed into airtight zip-lock bags and once the sampling campaign
was over, they were sent to Belgium for physicochemical soil analysis. Bulk density ($\rho_b$) was measured by oven drying (75°C
for 48 h) and weighing the soil samples. Soil porosity was derived from Eq. (1), assuming a particle density of 2.65 g cm$^{-3}$. pH
was measured by a potentiometric method using a pH-sensitive glass electrode, a standard reference electrode (HI 4222; Hanna
Instrument, Bedfordshire, UK), and a volumetric ratio soil:liquid of 1:5 for pH$_{water}$ (distilled water) and pH$_{KCl}$ (1M KCl). NO$_3^-$
and NH$_4^+$ content was determined colorimetrically (Auto Analyzer 3; Bran and Luebbe, Norderstedt, Germany) after
extractions performed with 1M KCl at room temperature and neutral pH. C and N concentrations (%C, %N), along with the
stable N isotope signatures ($\delta^{15}$N) of the soil samples, were determined at natural abundance by a Continuous Flow Element
Analyzer (Automated Nitrogen Carbon Analyzer), interfaced with an Isotope-Ratio Mass Spectrometer (Sercon 20-20; Sercon,



Cheshire, UK). Moreover, the soil samples taken at 5 and 20 cm depth were combined to produce one composite sample associated to each site, and by means of the method described by the International Organization for Standardization (ISO 11277:2009), soil texture was determined. The classification was made according to the classification system of the United States Department of Agriculture (USDA, 2017); and the soil class was determined based on the classification of FAO and
UNESCO: World Reference Base for Soil Resources (WRB) (FAO, 2007).

$$Porosity \; [\%] = \left(1 - \frac{\rho_b \; [g \; cm^{-3}]}{2.65 \; [g \; cm^{-3}]}\right) \cdot 100\% \qquad (1)$$

Daily measurements of soil moisture, expressed as water-filled pore space (WFPS), were taken per site at 5 and 20 cm depth
using soil moisture sensors (EC-5, Meter Environment, Pullman, Washington, USA) and data loggers (ProCheck, Meter Environment, Pullman, Washington, USA). Finally, soil temperature was determined daily for each measurement cycle and per chamber, by means of a thermometer inserted at 5 cm depth and approximately 10 cm from each chamber.

### 2.2.1 Soil-atmosphere exchange

12 h after the installation of the collars, the chambers were closed for a period of 1 h, and samples of 20 mL were taken with
disposable syringes from the headspace air of the chambers every 20 minutes: $T_1 = 0$, $T_2 = 20$, $T_3 = 40$ and $T_4 = 60$ min; $T_1$ or time-zero indicates the sample taken immediately after the chamber was closed. Moreover, prior to each sample collection, the syringe was flushed twice with air of the chamber to mix the chamber headspace and to avoid any possible stratification of them.

The 20 mL samples were injected in pre-evacuated 12 mL exetainer vials (over-pressurized), and once the sampling campaign was over, the samples were sent to Belgium for analysis by gas chromatography at Ghent University. For $CH_4$ and $CO_2$ analysis a gas chromatograph (Finnigan Trace GC Ultra; Thermo Electron Corporation, Milan, Italy) equipped with a flame ionization detector (FID) and a thermal conductivity detector (TCD) was used, respectively. For $N_2O$, another gas chromatograph equipped with an electron capture detector (ECD) (Shimadzu GC-14B; Shimadzu Corporation, Tokyo, Japan) was used.

### 2.2.2 $N_2O$ bulk isotopic composition

Two extra gas samples were taken for stable isotope analysis at the start ($T_1$) and at the end ($T_4$) of a chamber closure. These samples were taken only once per site and in only one of the chambers. For this, in addition to the small exetainer vials, pre-evacuated big serum vials (110 mL) were used to inject gas samples of 180 mL (over-pressured). At the end of the field campaign, the samples were transported to Switzerland (ETH Zurich) and analyzed for bulk $^{15}N$ measurement of $N_2O$ ($\delta$
$^{15}N^{Bulk}$) using a gas preparation unit (Trace Gas, Elementar, Manchester, UK) coupled to an Isotope Ratio Mass Spectrometer (IRMS) (IsoPrime100, Elementar, Manchester, UK). For measurement and calibration details see Verhoeven et al. (2019).





### 2.3 Data analysis

All statistical analyses were conducted in R Studio, version 3.5.2 (The R Core Team, 2019), and the statistical significance was reported at 95% confidence level ($P \leq 0.05$), unless otherwise stated.


Mean values with standard deviations (SD) per site and depth were calculated for the physicochemical soil properties. The fluxes for each gas ($CO_2$, $CH_4$ and $N_2O$) were calculated by means of linear regressions using the four consecutive measurements of each measurement cycle. The slope of the regressions represented the flux. Thus, following the ideal gas law, and considering the head space volume of the chamber and the chamber area, the net gas flux was calculated by Eq. (2) (Collier

et al., 2014; Dalal et al., 2008; Kutzbach et al., 2007):

$$F_c = \left(\frac{\Delta c \,[ppm]}{t \,[min]}\right) \cdot \left(\frac{P \,[atm]}{R \,[L\, atm\, mol^{-1}\, K^{-1}] \cdot T \,[K]}\right) \cdot (MW \,[g\, mol^{-1}]) \cdot \left(\frac{V_{ch} \,[L]}{A_{ch} \,[m^2]}\right) \quad (2)$$

where Fc corresponds to the net gas flux ($CO_2$, $CH_4$ or $N_2O$), $\Delta c/t$ is the rate of change of the gas concentration within the

chamber or the slope of the regression line [ppm min$^{-1}$ or µL L$^{-1}$ min$^{-1}$], P/RT corresponds to the ideal gas law used to convert concentration from volumetric to mass at normal temperature and pressure: P = absolute pressure (1 atm), R = gas law constant (0.08206 L atm mol-1 k-1), and T = temperature (293 K); MW is the molecular weight of the gas ($CO_2$-C and $CH_4$-C: 12.01 g mol-1, $N_2O$-N: 14.01 g mol-1), $V_{ch}$ is the headspace volume of the chamber, and $A_{ch}$ the area of the chamber. The goodness-of-fit was evaluated for every linear regression using the adjusted coefficient of determination ($R^2$), and time series

(concentration vs time) with $R^2 < 0.60$ were excluded from further analysis.

For the purpose of evaluating if there were any statistical differences at each site between the fluxes obtained in August and September, a one-way ANOVA was performed per site; verifying for each case the respective assumptions (i.e. equality of variances and normality). Moreover, a linear model was fitted using the sites as a factorial explanatory variable per gas flux

measurement to assess differences across sites, to estimate the effect sizes of the net fluxes, and to determine to which extent the variability of the net fluxes could be explained by these explanatories. For this, the validity of the model was evaluated through verification of assumptions of linearity, homoscedasticity (or equality of variances), and normality of the error terms; and due to the nature of the data in the measurements of $N_2O$, the $N_2O$ fluxes were log-transformed to homogenize variances.

In order to determine the physicochemical soil characteristics able to explain to the greatest extent the net fluxes of $CO_2$, $CH_4$ and $N_2O$ (response variable), a preliminary stepwise multiple linear regression was done for each soil gas. However, due to the low amount of data points (number of plots per altitude) and the small variability within plots, we averaged the measurements out, rather than to explicitly use separate measurements in a more complex linear mixed effect model. Therefore, we started with a full model that included all variables measured at 5 cm depth (predictors), and the average per plot and per





day of the fluxes measured only during the first month (August) (the soil samples for physicochemical soil properties were only taken in August). Then, using the "step" function of the "stats" package in R (The R Core Team, 2019) and the Akaike Information Criterion (AIC), a model was selected for each gas. Subsequently, the variance inflation factor (VIF) was calculated to avoid multicollinearity problems between predictors, and by means of the "vif" function from the "car" package in R (The R Core Team, 2019), along with a VIF threshold of < 3, the predictors with a higher VIF were excluded one by one.

Finally, a simple linear regression was carried out for each flux with the retained predictors, verifying in each case the validity of the model and the respective assumptions (i.e. linearity, homoscedasticity and normality of the error terms).

The soil isotopic signature of $N_2O$ was calculated using a two-source mixing model. Based on the conservation of mass depicted in Eq. (3) - where the atmospheric concentration of $N_2O$ ($C_a$) reflects the background atmospheric concentration of

the gas ($C_b$), plus the amount added by the source(s) ($C_s$) - and including the isotope ratios of each component (4), the soil isotopic signature of $N_2O$ - or $\delta^{15}N_s^{Bulk}$ - can be calculated by Eq. (5). However, due to the nature of the data (very low $N_2O$ concentrations), a minimum concentration difference of 20 ppb was defined as threshold to remove super low fluxes and thus, avoid larger uncertainties in the source calculation.


$$C_a = C_b + C_s \tag{3}$$

$$\delta^{15}N_a^{Bulk} \cdot C_a = \delta^{15}N_b^{Bulk} \cdot C_b + \delta^{15}N_s^{Bulk} \cdot C_s \tag{4}$$

$$\delta^{15}N_s^{Bulk} = \frac{\delta^{15}N_a^{Bulk} \cdot C_a - \delta^{15}N_b^{Bulk} \cdot C_b}{C_a - C_b} \tag{5}$$

Finally, in order to compare the fluxes of the three GHGs measured (i.e. $CO_2$, $CH_4$ and $N_2O$) and to determine the overall budget of the soils, the $CO_2$-eq emissions for each gas were calculated by means of Eq. (6), and by using a global warming

potential (GWP) of 1, 28 and 265 for $CO_2$, $CH_4$ and $N_2O$, respectively (IPCC, 2014a). Moreover, the total GHG budget at each site was obtained by summing the $CO_2$-eq emissions of each gas (Myhre et al., 2013).

$$CO_2 - eq_i \, [mg \, m^{-2}h^{-1}] = flux_i \, [mg_i \, m^{-2}h^{-1}] \cdot GWP_i \tag{6}$$

where *i* refers to $CO_2$, $CH_4$ or $N_2O$.

**3 Results**

**3.1 Physicochemical soil properties**

Soils are Andosols and the soil texture was classified (USDA) between loam and sandy loam at all sites (WRB; Table 2). All sites had a relatively acidic soil; $pH_{water}$ ranged from strong to medium acidic (4.61 - 5.69), with an increase in acidity with depth, except at P_3010 (Table 2). The most acidic soil was found at S_400 at 5 cm, although not significantly different from

M_1100 and C_2200; whereas the least acidic one at P_3010 at 5 cm depth, and only significantly different from M_1100.





Except for P_3010, $NO_3$-N concentrations were 2-4 times higher at 5 cm compared to 20 cm depth; the highest variability was observed at S_400, and in comparison to the other sites, P_3010 seems to be depleted in $NO_3$-N at both depths (0.8 – 3.6 µg g-1). In contrast, the highest concentration of $NH_4$-N was obtained at P_3010 at 20 cm, followed by S_400 at 5 cm. However, at all sites, $NH_4$-N concentrations at 5 cm were not significantly different from each other. Such as $NO_3$-N, $NH_4$-N also

decreased with depth, except at P_3010 where the increase at 20 cm with respect to 5 cm was almost doubling. Higher N contents were measured at 5 cm compared to 20 cm depth at all sites; and S_400 exhibited the highest content at both depths, 1.3-1.4 times higher than any other N percentage at the same depth, and even 4 times higher than any other N percentage at 20 cm depth. The C content showed a general decrease with depth at all sites, with the highest percentage at S_400 at 5 cm, and the lowest one at M_1100 at 20 cm. Higher $\delta^{15}N$ signatures were obtained at 20 cm compared to 5 cm depth; at S_400 the soil

was most enriched in $^{15}N$, and P_3010 showed the most depleted one.

Soil temperature decreased with altitude with a gradient of -4.2 °C per 1000 m, with no statistical difference between months. WFPS increased significantly with depth at all sites during both months (Fig. S2), except at C_2200 in September. The lowest WFPS at 5 cm depth was obtained at C_2200 (16.8%±2.5) and P_3010 (14.4%±0.3) in August and September, respectively,

whereas the highest one at M_1100 at 20 cm in both months (August: 75.9%±0.3; September: 71.9%±6.3).

### 3.2 Greenhouse gas fluxes

In general, all sites were sources of $CO_2$ (Fig. 1a, Table 3). Except for P_3010, mean $CO_2$ emissions were higher in September compared to August, but due to the high variability in the measurements, there was no significant difference between months at M_1100 and P_3010 ($P > 0.05$). The lowest and highest emissions were observed at C_2200 and P_3010, respectively, in

both months, and all sites were significant predictors and able to explain 56% of the variability of $CO_2$ emissions during the field campaign (Fig. 1a).

All mean $CH_4$ fluxes were negative, indicating a net flux from the atmosphere to the soil (Fig. 1b, Table 3). Although the mean $CH_4$ fluxes (except for P_3010) were higher in September compared to August, there was no significant difference ($P > 0.05$)

between months at any site. Moreover, the linear model performed for $CH_4$ only explained 3% of the variability.

Finally, the mean $N_2O$ fluxes showed a general negative trend with increasing altitude (Fig. 1c). A marked net $N_2O$ consumption was observed at the sites located at 2200 and 3010 m a.s.l., and besides these sites, M_1100 also acted as a net sink in September; however, there was no significant difference ($P > 0.05$) in any plot between months. The highest

consumption was observed in August at P_3010, while the highest emission was in September at M_1100 (Table 3). On the other hand, the fitted linear model explained 65% of the variability of the $N_2O$ fluxes during the field campaign.



Although only monthly average fluxes will be discussed, the large variability observed with most of the gas fluxes (Table 3 and Fig. 1) are the result of the spatial (i.e. differences in GHG fluxes between chambers) and temporal (i.e. differences in GHG fluxes per day) variability within each site.

### 3.3 Linear regressions with physicochemical soil characteristics

Changes in $pH_{water}$ and $NO_3$-N at 5 cm depth explained 83% of the temporal (daily measurements) and spatial (four sites) variability of $CO_2$ fluxes in August (Table 4). Both predictors were positively correlated with $CO_2$ emissions, thus, $pH_{water}$ ($P < 0.001$) and $NO_3$-N content ($P = 0.51$) were positively related to $CO_2$ fluxes. For $CH_4$ consumption, it was not possible to obtain a model since none of the predictors were retained during the stepwise selection. In case of $N_2O$ fluxes, bulk density ($P = 0.07$) and $pH_{water}$ ($P = 0.06$) at 5 cm depth explained 36% of the temporal and spatial variability in August. Although both predictors were close to the threshold ($P = 0.05$) and considered as non-significant, they indicate a negative correlation where every increase in bulk density and $pH_{water}$ lead to a decrease in $N_2O$ fluxes.

### 3.4 Isotopic signature of $N_2O$ ($\delta^{15}N_s^{Bulk}$)

$\delta^{15}N_s^{Bulk}$ ranged from -13.08 to 11.54‰, with the lowest and highest isotopic signature observed at S_400 (September) (Fig. 2, Table S1). During both months, $\delta^{15}N_s^{Bulk}$ values of M_1100, C_2200 and P_3010 exhibited $^{15}N$ enrichments, and all of them reflected chambers where negative fluxes were obtained i.e. consumption of $N_2O$ from the atmosphere to the soil (Table S1).

### 3.5 Soil GHG Budget

The average soil GHG balance indicates that all plots during August and September are considered as sources of GHGs, largely driven by $CO_2$ emissions (Fig. 3a), and with the highest compensation from $CH_4$ consumption (Fig. 3b). During both months, the highest and lowest $CO_2$-eq emissions were obtained at P_3010 (August: 499.6, September: 450.6 mg $CO_2$-eq m$^{-2}$ h$^{-1}$) and at C_2200 (August: 200.3, September: 248.3 mg $CO_2$-eq m$^{-2}$ h$^{-1}$), respectively; in both cases, $N_2O$ and $CH_4$ consumption resulted in a ~1% offset of the total $CO_2$-eq emissions.

### 4 Discussion

#### 4.1 GHG fluxes and correlations

##### 4.1.1 $CO_2$ fluxes

Across our study sites, P_3010 exhibited the highest soil $CO_2$ emissions (Fig. 1a and Table 3). Even though an increase in temperature (up to an optimum of ca. 50°C; Luo and Zhou, 2006a; Oertel et al., 2016), moisture-WFPS (up to an optimum 60%; Dalal and Allen, 2008; Luo and Zhou, 2006b) and $pH_{water}$ (up to an optimum 7; Oertel et al., 2016) generally lead to higher emissions of $CO_2$, P_3010 is the site with the lowest temperature and WFPS, but with the highest soil $pH_{water}$. Indeed,





under acid conditions, Sitaula et al. (1995) reported a 2 to 12 fold decrease in $CO_2$ emissions with decreasing $pH_{water}$ from 4.0 to 3.0; and Persson & Wiren (1989) indicated a decrease in $CO_2$ emissions of 83 and 78% with a decrease in $pH_{water}$ from 3.8 to 3.4 and 4.8 to 4.0, respectively. This is also supported by Luo & Zhou (2006a), Oertel et al. (2016), Reth et al., (2005) and

Wang et al., (2010) who have reported positive correlations between soil $pH_{water}$ and $CO_2$ fluxes. Hence, this dominant role of $pH_{water}$ in the overall $CO_2$ budget is also apparent across our study range. Nonetheless, shifts in C allocation could also give rise to shifts in $CO_2$ emissions, and thus support the increase observed in the site located at the highest altitude (P_3010). As such, an increase in fine root biomass is expected in tropical mountain forests compared to lowland forests. In fact, a study carried out in the South Ecuadorian Andes from 1050 to 3060 m a.s.l., indicates a positive linear correlation ($R^2 = 0.87$, $P =$

0.01) between fine root biomass and altitude, arguing that imbalances or limitations in resource (water and/or nutrients) availability at higher altitudes may be the cause (Leuschner et al., 2007). Consequently, the observed increase in $CO_2$ emissions at high altitude might be further driven by an increase in root biomass as the latter has been shown to be positively correlated with soil respiration (Han et al., 2007; Luo and Zhou, 2006a; Reth et al., 2005; Silver et al., 2005).

In contrast to P_3010, the low emissions observed at C_2200 could be attributed to (1) the lower WFPS, (2) the lower contents of C and N, and (3) the higher bulk density. Indeed, the lowest content of water was observed at this site in August at 5 cm depth, and exactly in this month, the lowest emissions of $CO_2$ were obtained. The low contents of C and N exhibited in C_2200 (indeed, the lowest from all the sites), could have hampered the $CO_2$ emissions, since an increase in C content normally leads to higher levels of respiration, and N itself is required for plants and soil microorganisms to grow (Dalal and Allen, 2008; Luo

and Zhou, 2006a; Oertel et al., 2016). In additions, this site also had the highest soil bulk density (i.e. lowest porosity), which could have led to a decrease in soil respiration either by a physical impediment for root growth or by a decrease in soil aeration for microbial activities (Dilustro et al., 2005; Luo and Zhou, 2006c, 2006a). For instance, Kowalenko et al. (1978) reported an almost double $CO_2$ emission from a clayey loamy soil (7.04 mg $CO_2$-C m$^{-2}$ h$^{-1}$) than from a sandy soil (3.75 mg $CO_2$-C m$^{-2}$ h$^{-1}$), and Dilustro et al. (Dilustro et al., 2005) reported an overall 1.5 increase in $CO_2$ emissions from a clayey soil (171.07 mg

$CO_2$-C m$^{-2}$ h$^{-1}$) with respect to a sandy one (117.07 mg $CO_2$-C m$^{-2}$ h$^{-1}$).

Finally, our measurements are enveloped by earlier work when framed in a broader and pantropical context (Table S3, Fig. 4). However, although a quantitative comparison is difficult to make due to differences in e.g. sampling durations (single seasons, whole year cycles or only specific dates), sampling frequencies (sub-daily, daily or monthly), measuring methods, intrinsic

site properties, etc., our fluxes are well below most of the $CO_2$ fluxes reported for South America, except for a study performed in Brazil at 130 m a.s.l. (17.4 mg $CO_2$-C m$^{-2}$ h$^{-1}$) (Verchot et al., 2000). Moreover, although it is not common to obtain higher fluxes at higher altitudes, only one study performed in Peru, between 2811 – 2962 m a.s.l., showed a flux similar in magnitude (dry season: 120 mg $CO_2$-C m$^{-2}$ h$^{-1}$, wet season: 170 mg $CO_2$-C m$^{-2}$ h$^{-1}$) (Jones et al., 2016) to our flux obtained at P_3010.



### 4.1.2 CH$_4$ fluxes

In general, all sites acted as net sinks for CH$_4$ (i.e. uptake of atmospheric CH$_4$ by soils). During the whole field campaign only one chamber at one site (S_400) and a specific date (08/09/2018) showed a net source of CH$_4$ (45.82 µg CH$_4$-C m-2 h$^{-1}$). However, there were no statistical differences between months. The mean CH$_4$ fluxes were quite similar even between sites, and there was no significant linear regression with the physicochemical soil characteristics able to explain the CH$_4$ fluxes. All sites exhibited indeed a high temporal and spatial variability (Fig. 1b and Table 3), but WFPS was measured in only one

chamber per site and generalized for the site itself. Moreover, due to the arrangement of the data for the linear regression (see section 2.3.), taking average-plot values instead of point-chamber values could have led to a loss of 'explaining power' in this case.

CH$_4$ consumption is generally higher compared to the majority of CH$_4$ fluxes reported in the literature (for tropical forest soils

in South America (Table S.3, Fig. 5). Only the fluxes observed in Peru between 1532–1769 m a.s.l. (dry season: -45.82 µg CH$_4$-C m$^{-2}$ h$^{-1}$) and 2811–2962 m a.s.l. (dry season: -66.71 µg CH$_4$-C m$^{-2}$ h$^{-1}$) (Jones et al., 2016) are of the same order of magnitude as our fluxes; keeping in mind that we measured at the end of the dry season. However, as mentioned previously for CO$_2$ emissions, this comparison must be treated with caution since there are different variables influencing the final results. Even so, except for one study carried out in Ecuador at 400 m a.s.l. (19.37 µg CH$_4$-C m$^{-2}$ h$^{-1}$), all fluxes depicted in Fig. 5 and

measured during a dry season (DS), represent a sink of CH$_4$, which is supported by the fact that humid tropical forests are responsible of 10% to 20% of the global soil sink for atmospheric CH$_4$ (Verchot et al., 2000).

### 4.1.3 N$_2$O fluxes

Only S_400 (both months) and M_1100 (September) (i.e. plots located at the lower locations) acted as net sources of N$_2$O (Fig. 1c, Table 3). In contrast, the lowest flux was observed in the plot located at the highest stratum (P_3010) during August

and September, which showed a general net consumption. The high emissions obtained at the lowest strata corroborate with literature data on lowland tropical forests (Butterbach-Bahl et al., 2004, 2013; Koehler et al., 2009) and are mainly attributed to (1) soil water content, (2) temperature, and (3) N availability.

Firstly, N$_2$O emissions in tropical forest soils are predominantly governed by WFPS, which influences microbial activity, soil

aeration and thus diffusion of N$_2$O out of the soil (Davidson et al., 2006; Werner et al., 2007). Ideally, the highest emissions via denitrification are observed between 60-80% WFPS, but they can vary between 50–80% or 60–90% depending on the soil physical properties (Butterbach-Bahl et al., 2013; Dalal and Allen, 2008; Davidson et al., 2006; Oertel et al., 2016). At lower percentages (optimum: 20%), nitrification takes place and although N$_2$O can be produced as well, it yields a higher potential for NO production (Davidson et al., 2006; Oertel et al., 2016). As a second main driver for N$_2$O emissions, and in comparison

to CO$_2$ emissions, denitrification is very sensitive to rising temperatures (Oertel et al., 2016). An increase in temperature leads





to an increase in soil respiration and thus to a depletion of $O_2$ concentrations, which is indeed a major driver in $N_2O$ emissions. Moreover, rising temperatures lead to a positive feedback in microbial metabolism; the stimulation of mineralization and nitrification processes induces an increase in the availability of substrates for denitrification, and thus to an increase in $N_2O$ emissions (Butterbach-Bahl et al., 2013; Sousa Neto et al., 2011). Hence, the high emissions observed at S_400 can also be

supported by the high temperatures observed at this site; which are indeed the highest from all sites (Fig. S1). Finally, the dependency of $N_2O$ emissions on WFPS and temperature is affected by substrate availability ($NO_3$) which was also the highest at S_400 (almost 2.5 times higher than the second highest content observed). High contents of $NO_3^-$ give an indication of an open or "leaky" N cycle with higher rates of mineralization, nitrification and thus $N_2O$ emissions (Davidson et al., 2006). Moreover, $NO_3^-$ is normally preferred as an electron acceptor over $N_2O$ and it can also inhibit the rate of $N_2O$ consumption to

$N_2$ (Dalal and Allen, 2008).

In contrast to the low elevation sites where net $N_2O$ emissions were observed, P_3010 presented the highest consumption (negative values, i.e. fluxes from the atmosphere to the soil), followed by C_2200. From 37 valid measurements only 1 resulted in net emission at P_3010 (range: -12.91 to 1.33 µg $N_2O$-N m$^{-2}$ h$^{-1}$), whereas from 36 measurements, 20 resulted in net

emissions at C_2200 (range: -11.10 to -0.31 µg $N_2O$-N m$^{-2}$ h$^{-1}$). Net $N_2O$ consumption is often related to N-limited ecosystems. Indeed, at low $NO_3^-$ concentrations, atmospheric or gaseous $N_2O$ may be the only electron acceptor left for denitrification (Chapui-Lardy et al., 2007; Goossens et al., 2001). Studies performed by Teh et al. (Teh et al., 2014) and Müller et al. (Müller et al., 2015) in the Southern Peruvian and Ecuadorian Andes, respectively, related the decrease in $N_2O$ emissions - and thus the potential for $N_2$ production in soils - at high elevations to differences in $NO_3^-$ availability.


In this case, P_3010 had the lowest content of $NO_3^-$, along with the lowest soil $\delta^{15}N$, which clearly reflects the shift towards a more closed N cycle at higher elevations (Bauters et al., 2017b). Moreover, this was the site with 1) the highest content of clay and hence more microsites for $N_2O$ reduction, 2) the lowest soil water content (% of WFPS) and hence more diffusion of atmospheric $N_2O$ into the soil, 3) the highest pH-value, which could have alleviated inhibition of the nitrous oxide reductase

at low pH, and 4) the highest $CO_2$ emissions observed, which could have indeed led to the development of anaerobic microsites where denitrification could have occurred (Chapui-Lardy et al., 2007). This is also valid for M_1100, where $N_2O$ consumption was observed in August (3 out of 7 valid measurements; range: -10.48 to 9.18 µg $N_2O$-N m$^{-2}$ h$^{-1}$), whereas $N_2O$ emission in September (5 out of 16 valid measurements; range: -9.67 to 94.42 µg $N_2O$-N m$^{-2}$ h$^{-1}$). Although samples of soils were not taken in September, this month represents the transition or the beginning of the wet season in the region. Thus, it is expected to have

an increase in available soil N due to the accumulation of litter during the dry season and the following but rapid mineralization after soil rewetting at the beginning of the rainy season, which could have indeed led to pulses of higher $N_2O$ emissions (Werner et al., 2007).





Contrary to $CO_2$ emissions, $pH_{water}$ negatively affected $N_2O$ emissions ($R^2 = 0.36$). This relation has been already reported
previously (Baggs et al., 2010; Chapui-Lardy et al., 2007; Wrage et al., 2001), and pH has been considered as a "master
variable" to predict N transformations (Baggs et al., 2010). As a second predictor, bulk density also exhibited a negative
correlation with $N_2O$ emissions. Reflecting the fact that higher values of bulk density are translated into less oxygen diffusion
and thus, more anaerobic sites ideal for the reduction of $N_2O$ to $N_2$. Likewise, Klefoth et al. (2014) observed a decrease in
$N_2O$ fluxes of about ~ 71% when soil bulk density increased by only 0.13 units.


Finally, in comparison to other studies done in tropical forest soils in South America (Table S.3, Fig. 6),  our $N_2O$ emissions
(i.e. S_400 in both months and M_1100 in September) are in the same range as other fluxes reported in e.g. Ecuador (400
m.a.sl.: 7.53 and 7.36 µg $N_2O$-N $m^{-2}$ $h^{-1}$) and Brazil (400 m a.s.l.: 10 µg $N_2O$-N $m^{-2}$ $h^{-1}$; 1000 m.a.sl.: 9.0 µg $N_2O$-N $m^{-2}$ $h^{-1}$)
(Sousa Neto et al., 2011). The fluxes at the highest strata, however, seem to be relatively unique within a broader and
pantropical context (Fig. 6). There are only two studies performed at comparable altitudes in South America; the first one in
the Southern Ecuadorian Andes (2000 m a.s.l.: 2.05±0.64; and 3000 m a.s.l.: 0.47±0.62 µg $N_2O$-N $m^{-2}$ $h^{-1}$; Müller et al., 2015),
and the second one in the Southern Peruvian Andes (1700 m a.s.l.: 40.83±9.58 (dry season), 6.67±5.24 (wet season); and 2700
m a.s.l.: 7.92±7.08 (dry season), 0.83±9.17 (wet season); Teh et al., 2014). But, even though $N_2O$ consumption was observed
in point measurements, all of them showed a net emission. From this, and based on previous observations, it seems that (1)
$N_2O$ studies in the tropics are biased toward lowland forests (Müller et al., 2015; Purbopuspito et al., 2006), and (2) our results
along with the low fluxes (and even negative data points) reported especially in the Andes, highlight the importance of a
probably unaccounted sink of $N_2O$ at high altitudes; keeping in mind that tropical montane forests represent 11% of the world's
tropical forests (Müller et al., 2015).

**4.2 Isotopic signature of $N_2O$ ($\delta^{15}N_s^{Bulk}$)**

Previous studies have indicated that during the reduction of $N_2O$ to $N_2$, $N_2O$-reductase fractionates against $^{15}N$ (Barford et al.,
1999; Butterbach-Bahl et al., 2013; Menyailo and Hungate, 2006; Pérez et al., 2000). Consequently, complete denitrification
i.e. consumption of $N_2O$, leads to a $^{15}N$ enrichment of the residual $N_2O$, and thus to higher $\delta^{15}N_s^{Bulk}$ values (Park et al., 2011)
relative to the atmospheric bulk $N_2O$ composition (6.3‰; (Harris et al., 2017)).This indeed is reflected in the enriched $\delta^{15}N_s^{Bulk}$
values measured during $N_2O$ consumption while the $N_2O$ bulk signature during $N_2O$ production was highly depleted compared
to that of atmospheric $N_2O$ (two samples taken in September at S_400) (Fig. 2; Table S1). This is also in line with Park et al.
(2011) and Pérez et al. (2000) who have attributed $\delta^{15}N_s^{Bulk}$ values between -22 and 2‰ in natural tropical forest soils to
denitrification. Therefore, in addition to the soil isotope signatures, the bulk $N_2O$ isotope signatures confirm the net $N_2O$
consumption at higher altitudes, and net $N_2O$ emission at lower altitudes, and rule out that our net consumption rates are due
to sampling artefacts.





### 4.3 Soil GHG Budget

The differences in fluxes for each GHG are clearly visualized in Fig. 3. The high $CO_2$ emissions observed at P_3010 give rise to the highest $CO_2$-eq emissions, and in terms of non-$CO_2$ GHG, this plot also exhibited the highest sink due to $CH_4$ and $N_2O$ consumption. However, it is important to mention that the calculated $CO_2$-eq emissions for $CO_2$ reflect only the impact of soil

emissions (heterotrophic and autotrophic respiration) on the soil GHG budget, excluding photosynthesis and aboveground respiration. Therefore, based on the fluxes here obtained, the upland soils from our study clearly show a marked sink of non-$CO_2$ GHG. Even so, besides this and the known potential for C sequestration in tropical forests, the marked $CH_4$ and $N_2O$ sinks observed in this case, enhances the importance and protection of tropical forests, not only in terms of biodiversity and ecosystem services, but also as means of mitigation options to curb global warming.

## 5 Conclusions

Overall, we found that our $CO_2$ fluxes are well below most of the $CO_2$ fluxes reported in literature for tropical forest soils, with an unusual but marked increase at the highest altitude, mainly explained by soil pH and root biomass. Moreover, our $CH_4$ uptake fluxes are among the highest in the tropics (i.e. highest consumption of atmospheric $CH_4$) and reiterate the role of humid tropical forest soils as a known $CH_4$ sink. Contrary to the net $N_2O$ emissions observed in the lowest strata, the net

consumption at higher elevation seems to be quite unique and reflects (1) the worldwide bias of $N_2O$ studies toward lowland forests, (2) the need of coupling environmental and physicochemical soil variables with microbial analyses, and (3) the importance of conserving upland forest for $N_2O$ consumption. This net $N_2O$ uptake was confirmed independently by soil and $N_2O$ [15]N isotope signatures. Finally, although an altitudinal gradient was selected to evaluate the potential effect of temperature - and other factors that co-vary with altitude - on the GHG budget of the forest soils, our results for $CO_2$, $CH_4$ and $N_2O$ fluxes

clearly reflect the "complex" interplay of different environmental controls and physicochemical soil characteristics rather than a generalized trend related to altitude.

**Supplementary information**

**Fig. S1.** Overview map with the location of the study areas.

**Fig. S2.** Monthly average soil temperature (°C)±standard deviations (SD).

**Fig. S3.** Monthly average water-filled pore space (WFPS)±standard deviations (SD).

**Table S1.** $\delta^{15}N_S^{Bulk}$ values and $N_2O$ fluxes.

**Table S.2.** Measured and estimated $CO_2$, $CH_4$ and $N_2O$ fluxes from tropical forest soils of South America.

**Author contributions**

M.Bauters, H.V., SB. and P.B. developed the project; P.L. and M. Bauters carried out the field work and analyzed the data. All authors contributed to the ideas presented and edited the manuscript.



**Competing interests**

The authors declare that they have no conflict of interest.

**Acknowledgments**

This research has been supported by Ghent University, the VLIR-UOS South Initiative COFOREC (EC2018SIN223A103).
We also thank Mindo Cloud Forest Foundation, El Cedral Ecolodge and the Escuela Politécnica Nacional del Ecuador, for the
logistic support in Ecuador. M. Barthel was supported through ETH Zurich core funding provided to Johan Six.

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





**Table 1.** Characteristics of the study areas Río Silanche (400 m a.s.l.; S_400), Milpe (1100 m a.s.l.; M_1100), El Cedral (2200 m a.s.l.; C_2200) and Peribuela (3010 m a.s.l.; P_3010), including mean annual precipitations (MAP) and mean annual temperatures (MAT) extracted from the Worldclim data set, using average monthly data from 1970-2000 with a spatial resolution of ~1 km$^2$ (Fick and R.J. Hijmans, 2017).

| Study area | Forest type | Coordinates | | Altitude (m a.s.l.) | MAP (mm) | MAT (°C) |
|---|---|---|---|---|---|---|
| | | Latitude | Longitude | | | |
| **S_400** | Lowland evergreen forest of Choco | 00°08'45.58'' N | 79°08'34.22'' W | 400 | 3633 | 23.0 |
| **M_1100** | Andean foothill evergreen | 00°02'07.17'' N | 78°51'59.72'' W | 1100 | 2856 | 21.1 |
| **C_2200** | Andean montane evergreen | 00°06'47.87'' N | 78°34'10.88'' W | 2200 | 1464 | 16.8 |
| **P_3010** | Upper montane evergreen | 00°22'27.35'' N | 78°18'0.36'' W | 3010 | 956 | 12.8 |

Note: the coordinates were taken at the center of the plots.




**Table 2.** Physicochemical soil properties of the study areas Río Silanche (400 m a.s.l.; S_400), Milpe (1100 m a.s.l.; M_1100), El Cedral (2200 m a.s.l.; C_2200) and Peribuela (3010 m a.s.l.; P_3010) at 5 and 20 cm depth, including mean values±standard deviation (SD) of bulk density (ρb), porosity, pH in water (pH$_{water}$) and KCl suspension (pH$_{KCl}$), NO$_3^-$ and NH$_4^+$ concentration, bulk nitrogen (N) and carbon (C) content, carbon-to-nitrogen ratio (C/N) and δ$^{15}$N signatures from samples of soil taken in August. Similar lowercase letters in superscript and next to some values within one row and per depth (5 and 20 cm) indicate no significant difference at $P < 0.05$ between sites (S_400, M_1100, C_2200 and P_3010).

| | S_400 | | M_1100 | | C_2200 | | P_3010 | |
|---|---|---|---|---|---|---|---|---|
| | **5 cm** | **20 cm** | **5 cm** | **20 cm** | **5 cm** | **20 cm** | **5 cm** | **20 cm** |
| **Soil class** | Andosol[1] | | Andosol[1] | | Andosol[1] | | Andosol[1] | |
| **Soil texture** | Loam | Loam | Sandy loam | Sandy loam | Sandy loam | Sandy loam | Loam | Loam |
| **Sand (%)** | 41.0 | 40.0 | 70.8 | 67.0 | 63.7 | 60.5 | 41.9 | 45.0 |
| **Silt%)** | 43.4 | 47.0 | 21.7 | 27.9 | 29.7 | 34.4 | 32.5 | 34.9 |
| **Clay (%)** | 15.6 | 13.1 | 7.6 | 5.0 | 6.6 | 5.2 | 25.6 | 20.1 |
| **ρb (g cm$^{-3}$)** | 0.43±0.15[b] | 0.58±0.07[b] | 0.62±0.09[a,b] | 0.86±0.12[a] | 0.70±0.11[a] | 0.92±0.05[a] | 0.62±0.09[a,b] | 0.81±0.06[a] |
| **Porosity (%)** | 83.8±5.5[a] | 78.0±2.5[a] | 76.5±3.3[a,b] | 67.7±4.7[b] | 73.7±4.1[b] | 65.4±2.0[b] | 76.7±3.4[a,b] | 69.6±2.1[b] |
| **pH$_{water}$** | 4.6±0.7[a,b] | 5.2±0.5 | 4.6±0.8[b] | 5.5±0.4 | 4.8±0.4[a,b] | 4.8±0.6 | 5.7±0.5[a] | 5.6±0.5 |
| **pH$_{KCl}$** | 4.4±0.2[b] | 4.9±0.3[a,b] | 4.5±0.2[b] | 5.0±0.0[a] | 4.5±0.1[b] | 4.6±0.0[b] | 5.1±0.2[a] | 4.9±0.2[a,b] |
| **NO$_3$-N (µg g-$^1$)[2]** | 71.9±39.5[a] | 35.7±29.5[a] | 23.1±15.9[b] | 6.7±7.7[a,b] | 30.6±19.4[a,b] | 7.3±4.3[a,b] | 0.8±0.3[b] | 3.6±7.1[b] |
| **NH$_4$-N (µg g-$^1$)[2]** | 34.3±14.8 | 27.9±16.1[a,b] | 22.6±4.0 | 11.9±2.4[b] | 26.5±16.0 | 18.8±4.9[b] | 22.9±11.3 | 40.4±13.5[a] |
| **N (%)** | 0.8±0.2 | 0.5±0.1[a] | 0.6±0.2 | 0.2±0.1[b] | 0.6±0.2 | 0.3±0.0[a,b] | 0.6±0.0 | 0.4±0.2[a,b] |
| **C (%)** | 8.9±2.4 | 4.0±1.0[a,b] | 7.1±1.8 | 2.4±0.7[b] | 6.6±1.7 | 3.3±0.4[a,b] | 8.6±0.5 | 4.8±1.5[a] |
| **C/N[3]** | 10.6±0.4[c] | 8.9±0.4[c] | 11.9±0.6[b] | 10.6±0.7[b] | 11.8±0.8[b] | 10.4±0.5[b] | 14.6±0.5[a] | 12.8±1.3[a] |
| **δ$^{15}$N (‰)[4]** | 6.2±0.5[a] | 8.6±0.9[a] | 6.0±0.8[a] | 6.7±0.8[b] | 4.0±1.2[b] | 4.8±0.5[c] | 3.7±0.6[b] | 4.2±0.4[c] |

Notes: mean values±SD were calculated from soil samples taken adjacent to each soil chamber (n = 5), except for soil texture, where composites for each site at 5 and 20 cm depth were made from the soil samples taken from each chamber. [1]Commonly known as *Andisol* in the USDA Soil Taxonomy; [2]expressed per gram of dry soil; [3]calculated by dividing C (%) by N (%) in each soil sample; and [4]expressed relative to the international standard AIR.





**Table 3.** Average measurements±standard deviations (SD) of $CO_2$, $CH_4$ and $N_2O$ fluxes at Río Silanche (400 m a.s.l.; S_400), Milpe (1100 m a.s.l.; M_1100), El Cedral (2200 m a.s.l.; C_2200) and Peribuela (3010 m a.s.l.; P_3010) per month.

| Month | Plot | Average $CO_2$ flux (mg C m$^{-2}$ h$^{-1}$) | Average $CH_4$ flux (µg C m$^{-2}$ h$^{-1}$) | Average $N_2O$ flux (µg N m$^{-2}$ h$^{-1}$) |
|---|---|---|---|---|
| **August** | S_400 | 68.4±18.2 | -63.2±22.9 | 11.1±18.1 |
| | M_1100 | 69.0±8.5 | -55.6±22.3 | -0.2±7.7 |
| | C_2200 | 55.3±12.1 | -57.5±22.9 | -0.3±0.9 |
| | P_3010 | 137.6±32.8 | -55.5±17.6 | -6.1±3.2 |
| **September** | S_400 | 95.3±19.9 | -63.4±31.7 | 3.7±6.1 |
| | M_1100 | 74.6±24.0 | -56.3±16.2 | 13.2±31.3 |
| | C_2200 | 68.6±24.8 | -74.4±25.0 | -0.3±2.8 |
| | P_3010 | 124.0±26.9 | -46.7±14.7 | -5.1±1.9 |

Note: flux values represent the mean of 5 chambers per site and per measurement week using the four-point time series and considering the constraint set to evaluate linearity in each measurement cycle ($R^2 > 0.60$).





**Table 4.** Retained predictors of multiple linear regressions for each flux gas ($CO_2$, $CH_4$ and $N_2O$) in August. ρb stands for bulk density and "_5" for properties measured at 5 cm depth. $R^2$ is the adjusted coefficient of determination, and $P$ the significance level of the model. In each case, the $R^2$ and $P$ values reflect the result of the multiple regressions done for each flux and considering all the retained predictors i.e. considering the predictors that are even not significant. 'ns', '*', '**', and '***'
represent the significant levels of each estimate at $P > 0.05$ (non-significant), $0.01 < P \leq 0.05$, $0.001 < P \leq 0.01$, and $P \leq 0.001$, respectively.

| Flux response | Predictor | Estimate | $P$ value | $R^2$ |
|---|---|---|---|---|
| **$CO_2$** | Intercept | -287.25 | | |
| | $pH_{water}\_5$*** | 73.97 | 3.21e-07 | 0.83 |
| | $NO_3^-$-$N\_5^{ns}$ | 0.129 | | |
| **$CH_4$** | - | - | - | - |
| **$N_2O$** | Intercept | 61.96 | | |
| | $\rho_b\_5^{ns}$ | -34.57 | 1.40e-02 | 0.36 |
| | $pH_{water}\_5^{ns}$ | -8.15 | | |





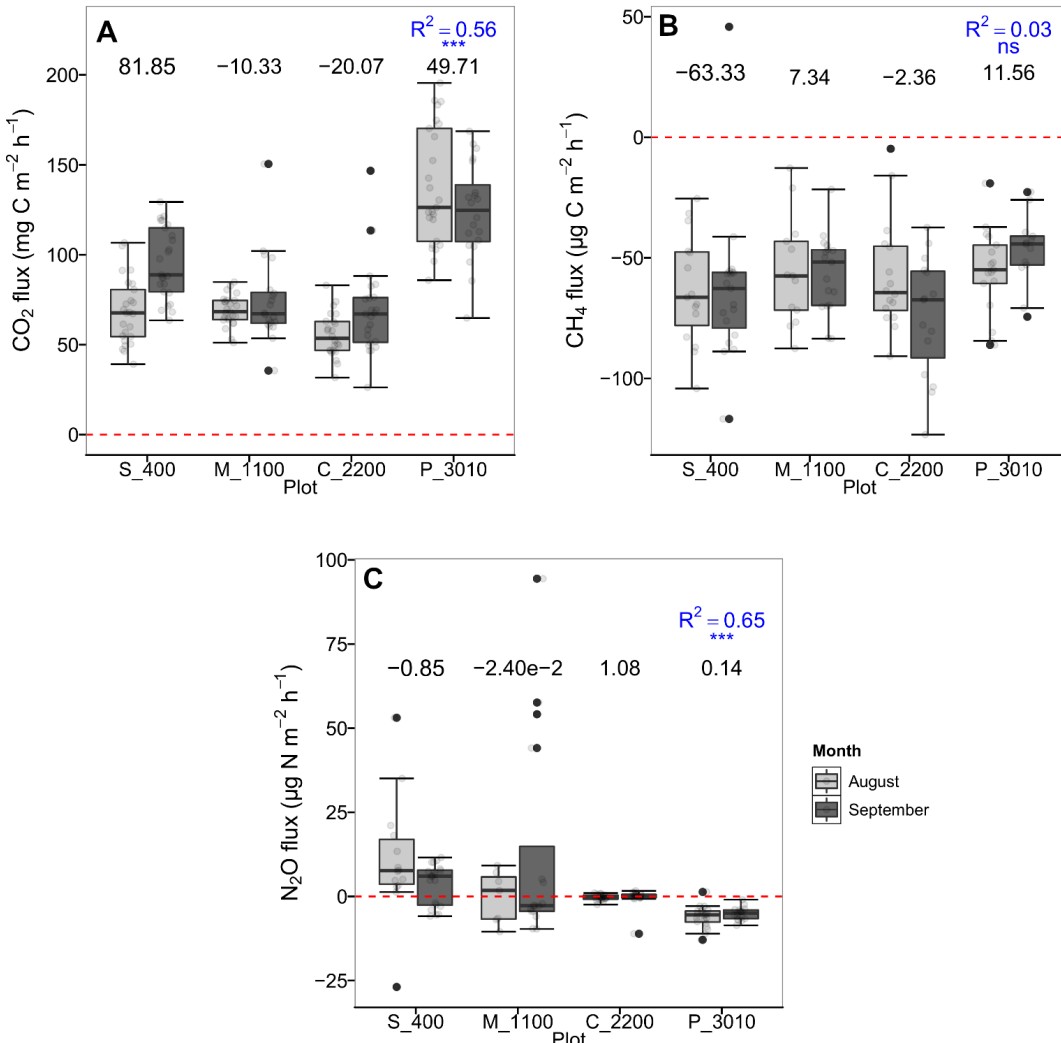

**Fig. 1.** A) $CO_2$ (mg C $m^{-2}$ $h^{-1}$), B) $CH_4$ (µg C $m^{-2}$ $h^{-1}$) and C) $N_2O$ (µg N $m^{-2}$ $h^{-1}$) fluxes per month at Río Silanche (400 m a.s.l.; S_400), Milpe (1100 m a.s.l.; M_1100), El Cedral (2200 m a.s.l.; C_2200) and Peribuela (3010 m a.s.l.; P_3010). Light gray boxplots indicate the fluxes in August 2018, whereas dark gray boxplots, the fluxes in September 2018. Light gray dots in each boxplot represent the measurements taken each day; and black dots, outliers of the respective site. Numbers above the

boxplots of each site and per flux gas correspond to the estimated coefficients obtained from a linear model to determine to which extent the sites can explain the variability of the net fluxes; S_400 is considered as the intercept; $R^2$ corresponds to the adjusted coefficient of determination of the respective model, and the symbols below it show the significance level of the overall effect at $P > 0.05$: 'ns' (non-significant), $0.01 < P \leq 0.05$: '*', $0.001 < P \leq 0.01$: '**', and $P \leq 0.001$: '***'. Note: $N_2O$ fluxes were $\log_{10}$-transformed.


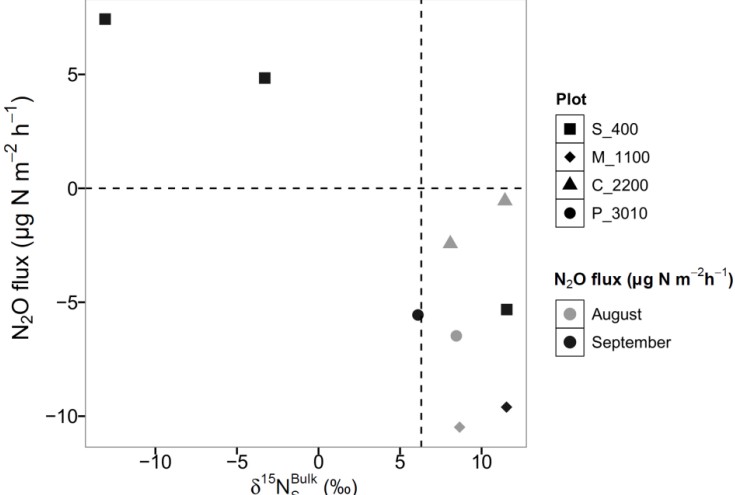

**Fig. 2.** N$_2$O fluxes plotted against the bulk isotopic signature of soil N$_2$O ($\delta^{15}N_S^{Bulk}$) for point samples taken at Río Silanche - squares (400 m a.s.l.; S_400), Milpe - diamonds (1100 m a.s.l.; M_1100), El Cedral - triangles (2200 m a.s.l.; C_2200) and

Peribuela - circles (3010 m a.s.l.; P_3010) during August (grey) and September (black). Note: the dotted x axis at 6.3‰ represents the atmospheric bulk N$_2$O composition (Harris et al., 2017), and $\delta^{15}N_S^{Bulk}$ values were calculated based on a two-source mixing model, considering a threshold of 20 ppb to exclude low fluxes and thus, avoid larger uncertainties in the source calculation.



**Fig. 3.** $CO_2$ equivalent ($CO_2$-eq) emissions at Río Silanche (400 m a.s.l.; S_400), Milpe (1100 m a.s.l.; M_1100), El Cedral (2200 m a.s.l.; C_2200) and Peribuela (3010 m a.s.l.; P_3010) during August and September 2018, and expressed as mg $CO_2$-eq $m^{-2}$ $h^{-1}$. Positive values indicate emission of GHGs whereas negative, consumption. Blue bars show the $CO_2$-eq emissions (-) of $CH_4$, green (+) of $CO_2$ and yellow (+ and -) of $N_2O$, using a global warming potential (GWP) of 1, 28 and 265, in each case and over a 100-year time horizon (Myhre et al., 2013). Figure B) is a zoom-in view of figure A).

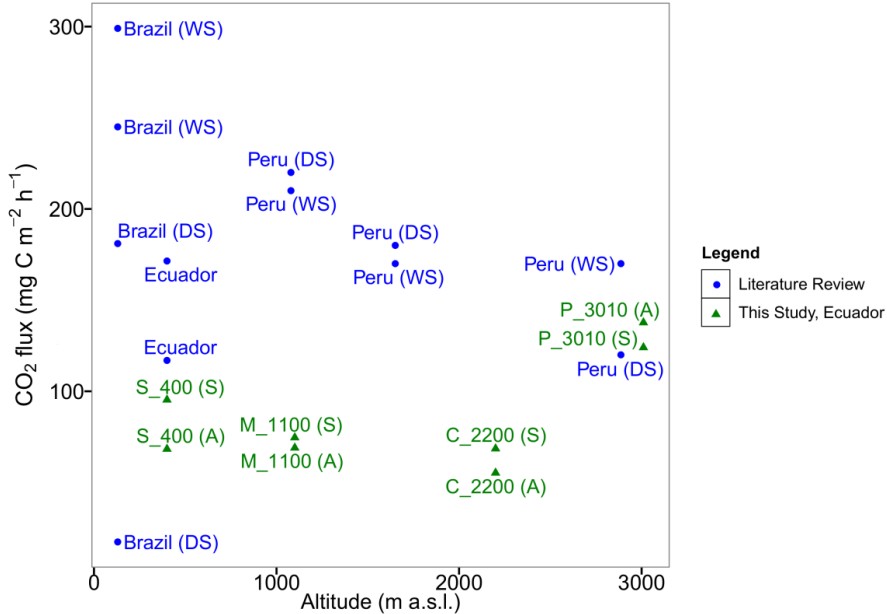

**Fig. 4.** Comparison of $CO_2$ fluxes with different studies. Blue dots: $CO_2$ fluxes reported in literature for South America (Table S.3); "DS" stands for fluxes taken specifically during a dry season, whereas "WS" during a wet season. Green triangles: $CO_2$ fluxes obtained in this study for Río Silanche (400 m a.s.l.; S_400), Milpe (1100 m a.s.l.; M_1100), El Cedral (2200 m a.s.l.; C_2200) and Peribuela (3010 m a.s.l.; P_3010); '(A)' denotes the fluxes obtained in August - end of the dry season in the region, and '(S)' the fluxes obtained in September – beginning of the rainy season.


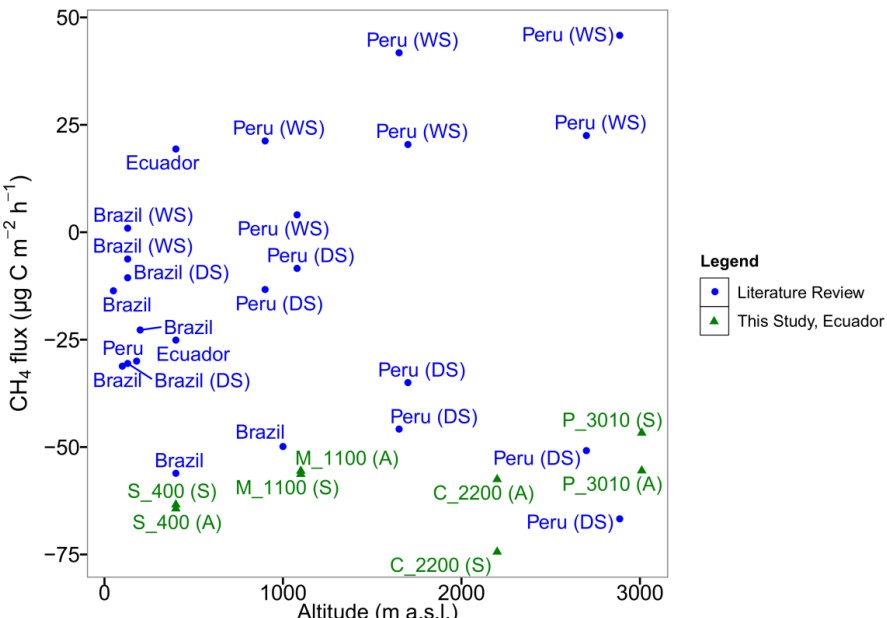

**Fig. 5.** Comparison of CH₄ fluxes with different studies. Blue dots: CH₄ fluxes reported in literature for South America (Table
S.3); "DS" stands for fluxes taken specifically during a dry season, whereas "WS" during a wet season. Green triangles: CH₄
fluxes obtained in this study for Río Silanche (400 m a.s.l.; S_400), Milpe (1100 m a.s.l.; M_1100), El Cedral (2200 m a.s.l.;
C_2200) and Peribuela (3010 m a.s.l.; P_3010); '(A)' denotes the fluxes obtained in August - end of the dry season in the
region, and '(S)' the fluxes obtained in September – beginning of the rainy season.




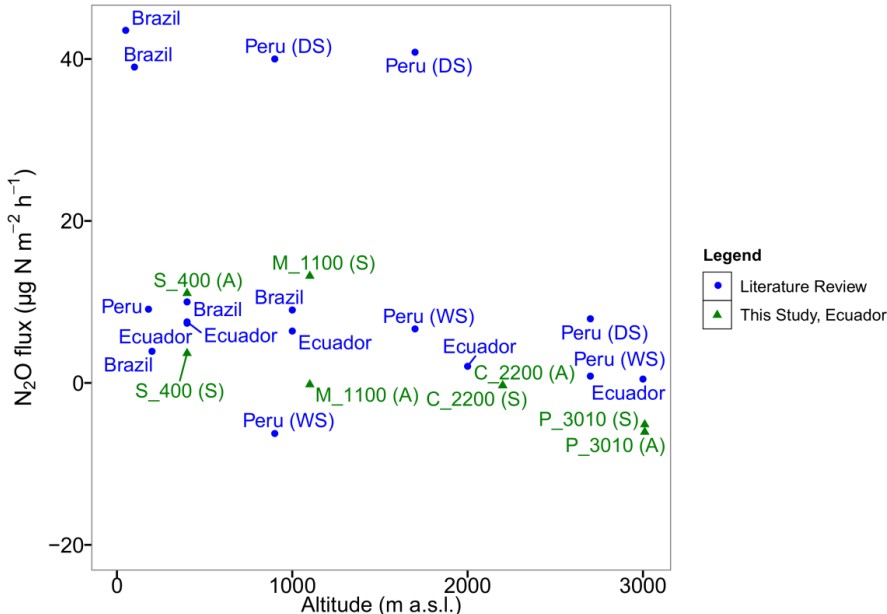

**Fig. 6.** Comparison of N$_2$O fluxes with different studies. Blue dots: N$_2$O fluxes reported in literature for South America (Table S.3); "DS" stands for fluxes taken specifically during a dry season, whereas "WS" during a wet season. Green triangles: N$_2$O fluxes obtained in this study for Río Silanche (400 m a.s.l.; S_400), Milpe (1100 m a.s.l.; M_1100), El Cedral (2200 m a.s.l.; C_2200) and Peribuela (3010 m a.s.l.; P_3010); '(A)' denotes the fluxes obtained in August - end of the dry season in the region, and '(S)' the fluxes obtained in September – beginning of the rainy season.