# Peer review of "Ideas and perspectives: patterns of soil CO2, CH4 and N2O fluxes along an altitudinal gradient - a pilot study from an Ecuadorian Neotropical montane forest"

_Biogeosciences, 2020_

## Referee Comment (RC1) · Paula Alejandra Lamprea Pineda et al. · 23 Apr 2020

That is a nicely written paper on soil CO2, CH4, N2O fluxes on a largely understudied tropical montane region in the Andes. My main concern, which is rather substantial, is that all interpretations are based on one week measurements (per strata) in August and again in September. I don't think that such a rather limited dataset allows for fundamental conclusions on site differences, magnitude of fluxes, correlations, temperature effects etc. as e.g. seasonal effects can't be considered and response ratios, e.g. to soil pH might change depending on sampling time and soil environmental conditions. I

am bit puzzled that data restrictions are not at all mentioned in the abstract, and conclusions. Conclusions: far reaching, but given the dataset, highly speculative as well as comparisons to existing datasets as done in the discussions. Sorry, but, what is needed here is to cook down the messages and implications within the paper as the dataset is very limited so that results, statistical interpretations etc. remain speculative. I do see the potential for an opinion paper, i.e. focusing on why such measurements are needed and how one might address the challenge to get tangible datasets for such remote regions and what can be achieved by short targeted campaigns (and as well important: what can't be achieved). For me the data are a starting point for a proposal for longer term measurements as would broaden the scientific knowledge on the contribution of these regions to the GHG balance and potential changes which might occur given the dynamic environmental changes occuring.

All the best

Klaus

Klaus
* * *

---

## Referee Comment (RC2) · Anonymous Referee #2 · 24 Apr 2020

In this study, the authors quantified soil fluxes of CO2, CH4) and N2O of four tropical forest sites located along an altitudinal gradient in northern Ecuador. This is an interesting study and we definitively need such dataset to complete our understanding of GHGs balance in tropical forest. Yet the spatial (one plot with 5 measurement points at each elevation and 4 different elevations : 20 measurement points) as well as the temporal coverage (one measurement per day during 5 consecutive days repeated twice : 10 measurements per sampling points only during the dry season) of the study are low. Moreover, collar insertion was done only 12h before the measurement. A longer

period between collar insertion and measurement is generally recommended in order to avoid effect of root death on $CO_2$ effluxes. These limitations are not mentioned in the abstract nor in the conclusion and can give the false impression that this study is presenting a larger dataset. Moreover, the review that is included in the last part of the paper is not complete, including only paper published before 2016 (see supplementary table). This review needs to be completed and the period of measurement (dry vs. wet) need to be specified. My conclusion is therefore that this paper is not suitable for publication in its current form. I nonetheless think that these data are worth being published but rather more as a short note presenting preliminary data.

―――――――――――――――

---

## Author Response (AR1)

**Reviewer # 1**

Klaus Butterbach-Bahl (Referee)

*"That is a nicely written paper on soil $CO_2$, $CH_4$, $N_2O$ fluxes on a largely understudied tropical montane region in the Andes. My main concern, which is rather substantial, is that all interpretations are based on one week measurements (per strata) in August and again in September. I don't think that such a rather limited dataset allows for fundamental conclusions on site differences, magnitude of fluxes, correlations, temperature effects etc. as e.g. seasonal effects can't be considered and response ratios, e.g. to soil pH might change depending on sampling time and soil environmental conditions. I am bit puzzled that data restrictions are not at all mentioned in the abstract, and conclusions. Conclusions: far reaching, but given the dataset, highly speculative as well as comparisons to existing datasets as done in the discussions. Sorry, but, what is needed here is to cook down the messages and implications within the paper as the dataset is very limited so that results, statistical interpretations etc. remain speculative. I do see the potential for an opinion paper, i.e. focusing on why such measurements are needed and how one might address the challenge to get tangible datasets for such remote regions and what can be achieved by short targeted campaigns (and as well important: what can't be achieved). For me the data are a starting point for a proposal for longer term measurements as would broaden the scientific knowledge on the contribution of these regions to the GHG balance and potential changes which might occur given the dynamic environmental changes occuring.*

*All the best*

*Klaus"*

**Reviewer # 2**

Anonymous Referee #2

*"In this study, the authors quantified soil fluxes of $CO_2$, $CH_4$ and $N_2O$ of four tropical forest sites located along an altitudinal gradient in northern Ecuador. This is an interesting study and we definitively need such dataset to complete our understanding of GHGs balance in tropical forest. Yet the spatial (one plot with 5 measurement points at each elevation and 4 different elevations: 20 measurement points) as well as the temporal coverage (one measurement per day during 5 consecutive days repeated twice: 10 measurements per sampling points only during the dry season) of the study are low. Moreover, collar insertion was done only 12h before the measurement. A longer period between collar insertion and measurement is generally recommended in order to avoid effect of root death on $CO_2$ effluxes. These limitations are not mentioned in the abstract nor in the conclusion and can give the false impression that this study is presenting a larger dataset. Moreover, the review that is included in the last part of the paper is not complete, including only paper published before 2016 (see supplementary table). This review needs to be completed and the period of measurement (dry vs. wet) need to be specified. My conclusion is therefore that this paper is not suitable for publication in its current form. I nonetheless think that these data are worth being published but rather more as a short note presenting preliminary data."*

**General response to reviewers:**

We would like to thank the two reviewers for their assessment of our manuscript. We acknowledge that this dataset is limited and agree with the reviewers that we should interpret the data likewise. Hence, we can review the manuscript for overstatements, and shorten the general manuscript. We are still convinced, however, that tropical montane forests are largely understudied in terms of GHG emissions [especially at higher elevations], asking for reporting – even with currently limited datasets. Moreover, the combination of the isotope analysis with a range in $N_2O$ fluxes (from net emission to net consumption, along the elevational gradient) is – in our opinion – novel and important. Additionally; only a few full soil GHG balances have been reported for tropical forests that range so strongly in environmental conditions and most of those also suffer from similar temporal constraints. This is in part due to the generally poor accessibility of these forests, especially at high elevation, hampering the deployment of novel portable analyzers and more automated systems to obtain a longer temporal coverage. Although the temporal coverage is indeed limited, the observation of a shift from a net positive to a net negative non-$CO_2$ GHG balance with increasing altitude is new and seems to be of interest to the readership of Biogeosciences. Therefore, we are willing to shorten and restructure the manuscript as suggested, as a shorter research paper, or in another manuscript type.

**List of relevant changes:**

In general, the manuscript has been restructured as an "ideas and perspective" piece. The length has been shortened. A better interpretation of the results has been carried out. Data limitations -mainly related to temporal and spatial coverage- have been clearly stated in the abstract and conclusions sections. We have focused on the importance of GHG studies in tropical regions, and as proposed by Reviewer #1, we have indicated the relevance of short-term campaigns, and we have provided insights for future and more detailed studies in tropical montane forests; explaining why those measurements are needed and how it might be possible to obtain tangible datasets in remote regions. Additionally, as indicated by Reviewer #2, the review included in the supplementary material ("Measured and estimated annual $CO_2$, $CH_4$ and $N_2O$ fluxes from tropical forest soils of South America") has been completed; a wider range of studies has been included (from 1983 to 2019), specifying in each case, the period of measurement.

- The title of the manuscript has been changed from "$CO_2$, $CH_4$ and $N_2O$ fluxes along an altitudinal gradient in the northern Ecuadorean Andes: $N_2O$ consumption at higher altitudes" to "Ideas and perspectives:  varying sources and sinks of $CO_2$, $CH_4$ and $N_2O$ along an altitudinal gradient - a pilot study from an Ecuadorian Neotropical forest"
- The title of each section has been renamed; thus, the introduction is divided into two sections: "1 The importance of tropical forests for GHG budgets" and "2 Altitudinal gradients as a biogeochemical open-air laboratory"; and the discussion of the results obtained are under a new section called: "3 What did we see in Ecuador?"
- The materials and methods, as well as results (sections 2 and 3 in the previous version) have been included in the supplementary information.
- Data restrictions (spatial and temporal coverage) are mentioned in the abstract and conclusions.
- The statistical differences on GHG fluxes between sites have been removed, as well as the linear regressions performed to determine the physicochemical soil characteristics able to explain the net GHG fluxes.
- The discussion of the results does not include anymore a comparison with the existing datasets.
- The conclusions section now includes insights for future studies in tropical regions.

---

## Author Response (AR2)

**General response to reviewers on the manuscript: "*Ideas and perspectives: patterns of soil $CO_2$, $CH_4$ and $N_2O$ fluxes along an altitudinal gradient - a pilot study from an Ecuadorian Neotropical montane forest*"**

Paula Alejandra Lamprea Pineda[1], Marijn Bauters[2, 3], Hans Verbeeck[3], Selene Baez[4], Matti Barthel[5] and Pascal Boeckx[2]

Firstly, we would like to thank the reviewers and the editor for reviewing and providing constructive comments to improve the manuscript.

Below, a point-by-point response to the reviews is shown, with the comments of the reviewers first, followed by our response in *italic* font. The manuscript has been adapted as noted below.

As a relevant change, it is important to mention that a flux correction has been carried out, considering the effect of local pressure and temperature. Previously, the fluxes had been calculated at normal temperature and pressure conditions (i.e. P: 1 atm, T: 293 K), and we feel that these pressure-corrected values are more correct than non pressure-corrected. However, the changes in the flux quantification is negligible and does not in any way change the message of this paper. In this sense, the GHG fluxes have been updated throughout the manuscript, including tables, graphs and supplementary material.

**Reviewer # 1**

Klaus Butterbach-Bahl (Referee)

Dear authors,

thanks for the revision and restructuring of the manuscript. Looks more convincing to me now. I still find the discussion of results on soil $CO_2$ fluxes highly speculative as no details on site, stand properties are provided. However, the point on $N_2O$ uptake, and the support of negative $N_2O$ flux data by evidence from isotopic signatures pointing towards $N_2O$ consumption is convincing and very interesting. I am surely supporting your lea for more research on tropical mountainous forest ecosystems under climate change, but this might need to go along as well with capacity building in the respective countries.

*The discussion on soil $CO_2$ fluxes have been shortened and the fact that we did not measure, nor estimated root biomass is clearly stated (see answer below for line Line 95-112). On the other hand, we agree that more research on tropical montane forests must go along with capacity building in the respective countries, thus, this has been clearly stated in the conclusions (L223).*

Here are a few additional points:

Line 37 while the argument is correct for $CH_4$ and $N_2O$, soil $CO_2$ fluxes are highly dependent on root (autotrophic) respiration too. Revise sentence (and see your line 40 too).

*Line 37 & 40 have been revised and restructured from: "In general, soil $CO_2$, $CH_4$ and $N_2O$ production or consumption depend on microbiological processes driven by a wide range of abiotic and biotic characteristics. The combination of these processes ultimately determines if a soil is a net source or sink of GHGs. Under aerobic conditions, $CO_2$ is emitted to the atmosphere by autotrophic and heterotrophic respiration (Dalal and Allen, 2008)" to "In general, soil $CO_2$ is produced mainly by root respiration, microbial respiration, litter decomposition, and oxidation of soil organic matter (Dalal and Allen, 2008). $CH_4$ is consumed by methanotrophic bacteria (Jang et al., 2006), however,*

*forest soils prone to inundation emit $CH_4$ by methanogenic microorganisms (Archaea domain). $N_2O$ is emitted through denitrification or a number of alternative pathways (e.g. nitrification, nitrifier-denitrification, chemodenitrification, etc. (Butterbach-Bahl et al., 2013; van Cleemput, 1998; Clough et al., 2017)), but can also be consumed during complete denitrification (Butterbach-Bahl et al., 2013)."*

Line 44 tropical forest soils

*Done*

Line 46 Provide a reference for these numbers/ estimates of NEE

*Reference provided*

Line 57 I would suggest to include Table S1 in the main text

*Although Table S1 gives an overview of all the studies that have been carried out on GHGs in South America, including it in the main text may deviate the main focus of the manuscript. The table is relatively long, and it would be more suitable for a review paper. Therefore, we refrain from including it to the main text.*

Lines 95-112 I find the discussion on the soil $CO_2$ emissions extremely speculative as there is no information on vegetation biomass, root biomass, vegetation type etc. Given the acknowledged importance of root respiration and the known high temporal variability of soil respiration, and the missing stand information, I think that little can be learned out of this and would suggest to further shorten this part.

*We agree on this. Therefore, the discussion on $CO_2$ emissions has been reduced and it has been clearly stated that root biomass was not measured nor estimated.*

Line 115 just mention here the number of observations day

*Done*

Line 139 check numbers for C_2200. If 20 fluxes should result in net emissions the range should have a positive value too

*Indeed, the range has been corrected.*

Line 144 typo with "-and"

*Done*

Line 154-158 see also Denk et al. 2017, The nitrogen cycle: A review of isotope effects and isotope modelling approaches. Soil Biol. Biochem. 105, 121-137

*Thanks for the suggestion, we have revised the paper and included Denk et al. 2017 in our discussion.*

Line 177 mention that root biomass was not measured nor estimated

*Done*

Line 195-200 Possibly also mention the importance for capacity building on biogeochemistry and GHG research in the target regions. Given that these regions will be severely hit by climate change, but so far, knowledge mainly comes from outside, such a network approach would also be helpful to

create a local knowledge base on how climate (and landuse change) might affect ecosystems (and people).

*We agree and included the suggestions in L223.*

**Reviewer # 2**

Anonymous Referee #2

This is an interesting work on soil GHG fluxes in an underrepresented region; it does not only provide a good (and short) dataset but also some insights into soil $N_2O$ uptake. I have had a look to the previous version and the comments of the reviewers and I agree with them that the experimental set up and the dataset fits better into a "preliminary data" paper. The paper is nicely and clearly written, and it is honest with regard to its limitations; at the same time, serves in suggesting ways forward for increasing our understanding on the patterns of GHG fluxes in tropical montane forest soils.

Probably one of the most relevant changes I suggest is with regard to the title. "Varying sources and sinks" seems to me that you look at different components within the forest, but this is not the case. I suggest to make explicit mention to the soil (e.g. –only a suggestion- "Ideas and perspectives: patterns of soil $CO_2$, $CH_4$ and $N_2O$ fluxes along …..") and probably to montane forests.

Further, I understand the rationale of using GWP to compare between GHGs, but I strongly doubt the usefulness of including $CO_2$ fluxes since, as you mentioned in the paper, it is only a part of the story and do not reflect inputs to the soil.

Finally, you highlight quite prominently the topographic position as a key driver controlling spatial variability (e.g. L32, L186); while this is true, other sources of variability may be equally important (e.g. degradation/forest management, hydrological status at the catchment level, soil types, exposition and associated microclimate) and should be taken into account when proposing "broader studies".

For the rest, I only have a couple of minor comments, which are depicted below.

*We thank the Referee for this positive assessment. We agree that the title can give a wrong perception about the scope of the manuscript, therefore, the suggestion has been accepted indicating clearly that the research was carried out on "montane" tropical forests.*

*We agree with the Referee on the usefulness of including the $CO_2$-eq emissions, since it does not add any extra information to the manuscript, we have decided to remove it from the text.*

*Moreover, we agree that besides topographic position there are more drivers controlling GHG fluxes, therefore we have included them on L197-209 as well as in the abstract L32.*

L41: Remove "(anaerobic)"

*Done*

L44-46. I am missing a citation here, so I can´t check on my own. I guess this is the C sink of the forest, and not of the soil; since you previously referred to the soil, it is misleading, unless you are able to infer what proportion is stored in the vegetation and in the soil (or below- and aboveground).

*The citation has been added. The number indeed refers to the C sink of the forest, and since it is not possible to infer the proportion stored in the vegetation and in the soil, it has been clearly stated that the number refers to the below and aboveground sink.*

L70: Probably merge the two paragraphs.

*Done*

L75: Remove "more" before "severe". Remove "even though forests can be managed to mitigate climate change as well", it is not really relevant here. Eventually, you can state that it is important to understand the feedbacks to come up with appropriate forest management options to mitigate climate change.

*Done*

L93: I suggest to change "budget" by "fluxes".

*Done*

L102: Zimmermann et al 2009(Eur J Soil Sci, doi: 10.1111/j.1365-2389.2009.01175.x) worked also on an altitudinal gradient in the Andes and pointed towards changing C allocation patterns. The paper might be also interesting for the summary table (I was not involved in this work).

*Thanks for the recommendation. This paper has been included in the summary table.*

L119: Remove "on the other hand". Reformulate the second sentence. The lowest flux is the closest to 0, and this is not what you want to point out.

*"On the other hand" has been removed, and the statement has been reformulated to: "Only S_400 and M_1100 (both months) (i.e. plots located at the lower locations) acted as net sources of $N_2O$ (Fig. 1c and Table 1). Whereas the plots located at the highest stratum (P_3010 & C_2200) showed a general net $N_2O$ consumption during August and September." It is important to mention that due to the flux correction (pressure and temperature), the data changed slightly and M_1100 in August showed a net emission of $N_2O$ ($0.8 \pm 6.9$ µg N m$^{-2}$ h$^{-1}$) instead of a net sink ($-0.2 \pm 7.7$ µg N m$^{-2}$ h$^{-1}$) as it was reported before.*

L139: The range is wrong, I think, it should go from negative to positive.

*The range has been revised and updated with the corrected fluxes.*

L143: This is also similar to what Gerschlauer (Biogeosciences) found in a gradient in the Kilimanjaro. I don´t remember if they found relationships between $N_2O$ fluxes and the isotopic signature in the soil, but may be worth looking at it (not involved either in the work).

*This is indeed similar. No measurements regarding $N_2O$ fluxes were carried out, but the authors mention that soil $\delta^{15}N$ values suggest the tightest N cycling at high elevations (> 3000 m a.s.l.). Although they clearly claim that a conclusion about the nature for the N cycle (open or closed) should be made considering other processes as well (e.g. soil nitrate leaching). Nevertheless, we have cited this paper in the respective line.*

L152: Probably "support" or similar, rather than "confirm"

*Agreed, support fits better the statement.*

L186: Remove "For instance"

*Done*

L194: Why bi-weekly? To me, the sampling should cover seasonal fluctuations (dry vs. wet season), but also at a finer scale.

*The main message of L194 was that long-term data that covers season fluctuations (i.e. dry vs wet) is needed, although "bi-weekly" was mentioned to give an example of the sampling intervals, we agree that sampling at a finer scale is important, as stated in line L169, thus, "bi-weekly" has been removed.*

Fig 2: Note: "the dotted vertical line" instead of "the dotted x axis"

*Done*

Fig 3: Even if the three GHG are represented using $CO_2$-eq, consider an axe-break instead on the zoom-in view.

*As stated before, the section on $CO_2$-eq emissions has been removed.*

Table S1. This is a great compilation of information. However, I find difficult to compare the studies. I suggest to add more columns, in order to separate between measured and estimated annual, and also a field with comments/period of measurement). I also suggest not to use the publication as a basis, but the site (e.g. Garcia Montiel et al 2004 measured on different sites, so include a line for each, with the same reference). For me, manual and dynamic chambers are not mutually exclusive. Do you rather mean manual vs. automated? In any case, probably the frequency is more important than the method per se (e.g. manual chambers can have daily, or monthly frequency).

*Extra columns to separate between measured and estimated annual fluxes has been added for each GHG. However, and extra column with comments/period of measurement was not added since the period of measurement sometimes differs between GHG and/or conditions (e.g. dry season vs wet season). Moreover, as suggested, a lined has been added when in the same study different sites were evaluated. Finally, the distinction between manual vs dynamic chambers has been removed, although the latter referred to automated; the frequency is indeed more important, but not always mention in the papers.*

[revised manuscript text omitted]

¶

---

## Author Response (AR3)

**Response to the associate editor on the manuscript: "*Ideas and perspectives: patterns of soil CO₂, CH₄ and N₂O fluxes along an altitudinal gradient - a pilot study from an Ecuadorian Neotropical montane forest*"**

Paula Alejandra Lamprea Pineda[1], Marijn Bauters[2, 3], Hans Verbeeck[3], Selene Baez[4], Matti Barthel[5], Samuel Bodé[2] and Pascal Boeckx[2]

We would like to thank the associate editor for the extraordinary revision of our manuscript after we found an inconsistency in the $^{15}$N data. Since the analytical model employed (two-source mixing model) does not apply for our experimental conditions, and it is not clear how this might have consequences for the data interpretation, we have decided to leave out this section from the manuscript. The specific problem lies with the linear assumptions of the two-source mixing models: a constant atmospheric pool with constant $^{15}$N signature, mixing with a sustained N-source with a constant $^{15}$N signature. As the chamber headspace is in fact varying in concentration and $^{15}$N signature, as the N₂O consumption takes place, these conditions are not linear. We – unfortunately – only figured this out after the final acceptance of the paper. However, it seems incorrect to leave this in, as the consequence of the violation of these assumptions are unclear. After internal discussion, a much more complex model including a two-source mixing model, but also a Rayleigh-type equation to simulate both gross N₂O production and consumption would be needed to truly constrain the fluxes in a correct way. However, we do not have these data (we would need much more timepoints, and at least also $^{15}$N-NO₃, along with considerable analytical model development).

As stated before, by omitting these data, the message of the manuscript does not change and the discussion and thus, conclusions obtained still remain. However, a small discussion has been included to indicate the analytical advances needed to disentangle gross consumption and production of N₂O (see L152-165 final version).

On the other hand, regarding the following comment of the editor:
*One very minor suggestion which can be addressed during the proofing stage: I suggest to split the sentence L121-125 in two parts, e.g. by starting a new sentence on L124 ("Therefore, the observed..."). Right now, the sentence is ~80 words long, which seems a bit much of a good thing.*
L121-125 has been revised and shortened.

An updated version of the manuscript and supplementary information with track changes is added below:

[revised manuscript text omitted]
) | Measured $CO_2$ flux (g $CO_2$-C $m^{-2}h^{-1}$) and period of measurement | Estimated annual $CO_2$ flux (Mg $CO_2$-C $ha^{-1}y^{-1}$) | Measured $CH_4$ flux (µg $CH_4$-C $m^{-2}h^{-1}$) and period of measurement | Estimated annual $CH_4$ flux (kg $CH_4$-C $ha^{-1}y^{-1}$) | Measured $N_2O$ flux (µg $N_2O$-N $m^{-2}h^{-1}$) and period of measurement | Estimated annual $N_2O$ flux (kg $N_2O$-N $ha^{-1}y^{-1}$) | Reference |
|---|---|---|---|---|---|---|---|---|---|
| **Brazil** | 50 | 2000 - | N.R. | N.R. | -11.5 April 1983 | N.R. | 43.6 April 1983 | N.R. | (Keller et al., 1983) |
| **Brazil** | 50 | 2000 - | N.R. | N.R. | 0.7; -22.3 December 1983, and March 1984 | N.R. | 13.1; 31.0 December 1983, and March 1984 | N.R. | (Keller et al., 1986) |
| **Brazil** | 54 | 2200 - | N.R. | N.R. | N.R. | N.R. | 15-35 April 1987-April 1988 | 1.9 | (Luizão et al., 1989) |
| **Brazil** | 130 | 1750 - | Dry season: 240±20 November 1992[3]; Wet season: 290±20 May 1992[3] | N.R. | N.R. | N.R. | N.R. | N.R. | (Davidson and Trumbore, 1995) |
| **Brazil** | 150 | 2200 25.5 | N.R. | N.R. | -3.42 to -5.93 kg $CH_4$-C $ha^{-1}y^{-1}$ June 1992-December 1993[1] | | N.R. | N.R. | (Steudler et al., 1996) |
| **Brazil** | 130 | 1850 N.R. | N.R. | N.R. | N.R. | N.R. | *Fazenda Vitória – Primary forest* Dry season: 10.4±0.8 Wet season: 52.3±4 February 1995-May 1996 | 2.43 | (Verchot et al., 1999) |
| | | | | | | | *Fazenda Vitória – Secondary forest* Dry season: 6.9±1.1 Wet season: 16.2±1.3 February 1995-May 1996 | 0.94 | |

| | | | | | | | | | |
|---|---|---|---|---|---|---|---|---|---|
| | | | N.R. | N.R. | N.R. | N.R. | *Fazenda São José* 5.4±1.6 July 1996 | N.R. | |
| | | | N.R. | N.R. | N.R. | N.R. | *Fazenda São Sebastião* 2.0±0.7 July 1996 | N.R. | |
| **Brazil** | 130 | 1800 N.R. | N.R. | 20 | N.R. | N.R. | N.R. | N.R. | (Davidson et al., 2000) |
| | 130 | 1800 N.R. | N.R. | 18 | N.R. | N.R. | N.R. | N.R. | |
| **Brazil** | 130 | 1850 N.R. | Dry season: 181±9 Wet season: 299±14 April 1995-May 1996 | 20 | Dry season: -30.6±6.6 Wet season: 0.9±6.6 April 1995-May 1996 | -1.6 | N.R. | N.R. | (Verchot et al., 2000) |
| | 130 (Secondary forest since 1976) | 1850 N.R. | Dry season: 174±10 Wet season: 245±10 April 1995-May 1996 | 17.9 | Dry season: -10.6±5.6 Wet season: -6.2±2.5 April 1995-May 1996 | -0.7 | N.R. | N.R. | |
| **Brazil** | 145 | 2200 25.6 | N.R. | N.R. | N.R. | N.R. | 1.94±0.22 kg $N_2O$-N ha$^{-2}$y$^{-1}$ June 1992-January 1996 | | (Melillo et al., 2001) |
| **Brazil** | 150 | 2270 18.8-25.6 | N.R. | N.R. | N.R. | N.R. | Dry season: 1.3 August-September 1998 Wet season: 76.4, February-March 1998; 67.0 March 1999 | N.R. | (Garcia-Montiel et al., 2001) |
| **Brazil** | 150 | 2270 25.6 | 129.21± 31.93 November 2001 | N.R. | N.R. | N.R. | 13.13± 3.75 November 2001 | N.R. | (Garcia-Montiel et al., 2003) |
| **Brazil** | 200 | 2000 25 | N.R.[2] | N.R. | N.R.[2] | N.R. | 185 (clay soil), 15 (sandy soil) June-August 2000 | N.R. | (Varner et al., 2003) |
| **Brazil** | 150 | 2090-2270 | *Nova Vida 1* 148.1±9.7 | N.R. | N.R. | N.R. | *Nova Vida 1* 27.3±4.6 | N.R. | |

| | | | | | | | | | |
|---|---|---|---|---|---|---|---|---|---|
| | | 18.8-25.6 | 1992-1999 | | | | 1992-1999 | | |
| | | | *Nova Vida 2* 155.5±18.5 1992-1993 | N.R. | N.R. | N.R. | *Nova Vida 2* 19.2±7.0 1992-1993 | N.R. | |
| | | | *Proto Velho* 163.92 October 1993 and March 1994 | N.R. | N.R. | N.R. | *Proto Velho* 25.2 October 1993 and March 1994 | N.R. | |
| | | | *Jamari* 151.09 October 1993 and March 1994 | N.R. | N.R. | N.R. | *Jamari* 34.5 October 1993 and March 1994 | N.R. | (Garcia-Montiel et al., 2004) |
| | | | *Cacaulândia* 159.01 October 1993 and March 1994 | N.R. | N.R. | N.R. | *Cacaulândia* 34.3 October 1993 and March 1994 | N.R. | |
| | | | *Ouro Preto* 144.7 October 1993 and March 1994 | N.R. | N.R. | N.R. | *Ouro Preto* 30.5 October 1993 and March 1994 | N.R. | |
| | | | *Vilhena* 161.4 October 1993 and March 1994 | N.R. | N.R. | N.R. | *Vilhena* 28.1 October 1993 and March 1994 | N.R. | |
| **Brazil** | 200 | 2000 21 - 23 | 10.0±0.9 Mg $CO_2$-C ha$^{-1}$y$^{-1}$ September 1998-December 2002[1] | | -0.8±0.7 kg $CH_4$-C ha$^{-1}$y$^{-1}$ September 1998-December 2002[1] | | 2.6±1.0 kg $N_2O$-N ha$^{-2}$y$^{-1}$ September 1998-December 2002[1] | | (Davidson et al., 2004) |
| **Brazil** | 200 | 2000 25 | 138.4±4.3 (clay soil), 160.0±8.6 (sandy soil) June 2000-May 2001 | N.R. | -11.2±4.4 (clay soil), N.R. (sandy soil) June 2000-July 2011 | N.R. | 130±10 (clay soil), 14±2 (sandy soil) June 2000-July 2011 | N.R. | (Silver et al., 2005) |
| **Brazil** | 200 | 2000 21 - 23 | 12.8±1.0 Mg $CO_2$-C ha$^{-1}$y$^{-1}$ September 1998-April 2005[1] | | -1.2±0.7 kg $CH_4$-C ha$^{-1}$y$^{-1}$ September 1998-April 2005[1] | | 2.1±0.7 kg $N_2O$-N ha$^{-2}$y$^{-1}$ September 1998-April 2005[1] | | (Davidson et al., 2008) |
| **Brazil** | 100 | 3050 | N.R. | 13.3 | -2.7±0.5 kg $CH_4$-C ha$^{-1}$y$^{-1}$ | | 3.4±0.4 kg $N_2O$-N ha$^{-2}$y$^{-1}$ | | |

| Country | | | CO₂ | | CH₄ | | N₂O | | Reference |
|---|---|---|---|---|---|---|---|---|---|
| | | 19.1 - 25.5 | | | September 2006-August 2007[1] | | September 2006-August 2007[1] | | |
| | 400 | 3050 / 19.1 - 25.5 | N.R. | 13.6 | -4.9±8.0 kg CH₄-C ha⁻¹y⁻¹ September 2006-August 2007[1] | | 0.9±0.1 kg N₂O-N ha⁻²y⁻¹ September 2006-August 2007[1] | | (Sousa Neto et al., 2011) |
| | 1000 | 2300 / 19.1 - 25.5 | N.R. | 12.9 | -4.4±0.3 kg CH₄-C ha⁻¹y⁻¹ September 2006-August 2007[1] | | 0.8±0.2 kg N₂O-N ha⁻²y⁻¹ September 2006-August 2007[1] | | |
| **Ecuador** | 400 | 4500 / - | 171.6 August 1984 | N.R. | 19.4 August 1984 | N.R. | 7.5 August 1984 | N.R. | (Keller et al., 1986) |
| | 400 (Secondary forest of 5-10 years old) | 4500 / - | 117.0 August 1984 | N.R. | -25.1 August 1984 | N.R. | 7.4 August 1984 | N.R. | |
| **Ecuador** | 1000 | 2230 / 19.4 | N.R. | N.R. | N.R. | N.R. | 0.2±0.1 (lower slope), 0.3±0.1 (mid-slope), 0.4±0.1 (ridge) kg N₂O-N ha⁻²y⁻¹ May 2008-May 2009[1] | | (Wolf et al., 2011) |
| **Ecuador** | 2000 | 1950 / 15.7 | N.R. | N.R. | N.R. | N.R. | 1.3±0.2 (lower slope), 0.3±0.1 (mid-slope), 0.1±0.1 (ridge) kg N₂O-N ha⁻²y⁻¹ May 2008-May 2009[1] | | |
| | 3000 | 4500 / 9.4 | N.R. | N.R. | N.R. | N.R. | 1.1±0.1 (lower slope), 0.1±0.1 (mid-slope), -0.05±0.1 (ridge) kg N₂O-N ha⁻²y⁻¹ May 2008-May 2009[1] | | |
| | 1000 | 2230 / 19.4 | 10.3±0.8 (lower slope), 10.3±0.1 (mid-slope), 9.8±0.9 (ridge) Mg CO₂-C ha⁻¹y⁻¹ May 2008-May 2009[1] | | -5.5±0.7 (lower slope), -5.4±0.9 (mid-slope), -5.9±1.0 (ridge) kg CH₄-C ha⁻¹y⁻¹ May 2008-May 2009[1] | | N.R. | N.R. | (Wolf et al., 2012) |
| **Ecuador** | 2000 | 1950 / 15.7 | 8.8±0.4 (lower slope), 7.6±0.6 (mid-slope), 6.7±0.7 (ridge) Mg CO₂-C ha⁻¹y⁻¹ May 2008-May 2009[1] | | -2.3±0.3 (lower slope), -4.3±0.9 (mid-slope), -2.7±0.3 (ridge) kg CH₄-C ha⁻¹y⁻¹ May 2008-May 2009[1] | | N.R. | N.R. | |
| | 3000 | 4500 / 9.4 | 6.4±0.4 (lower slope), 5.7±0.7 (mid-slope), 3.7±0.5 (ridge) Mg CO₂-C ha⁻¹y⁻¹ May 2008-May 2009[1] | | -0.6±1.2 (lower slope), -1.6±0.4 (mid-slope), -1.0±0.1 (ridge) kg CH₄-C ha⁻¹y⁻¹ May 2008-May 2009[1] | | N.R. | N.R. | |
| **Ecuador** | 1000 | 2230 / 19.4 | N.R. | N.R. | N.R. | N.R. | 0.2±0.1 (2008), 0.5±0.1 (2009) kg N₂O-N ha⁻²y⁻¹ January 2008-September 2009[1] | | (Martinson et al., 2013) |

| Country | | | | | | | | | Reference |
|---|---|---|---|---|---|---|---|---|---|
| | 2000 | 1950 / 15.7 | N.R. | N.R. | N.R. | N.R. | 0.2± 0.03 (2008), 0.1±0.2 (2009) kg $N_2O$-N ha$^{-2}$y$^{-1}$ January 2008-September 2009 | | |
| | 3000 | 4500 / 9.4 | N.R. | N.R. | N.R. | N.R. | -0.03± 0.1 (2008), -0.3±0.3 (2009) kg $N_2O$-N ha$^{-2}$y$^{-1}$ January 2008-September 2009 | | |
| Ecuador | 1000 | 2230 / 19.4 | N.R. | N.R. | N.R. | N.R. | 0.57±0.26 kg $N_2O$-N ha$^{-1}$y$^{-1}$ November 2010-August 2012[1] | | |
| | 2000 | 1950 / 15.4 | N.R. | N.R. | N.R. | N.R. | 0.17±0.06 kg $N_2O$-N ha$^{-1}$y$^{-1}$ November 2010-August 2012[1] | | (Müller et al., 2015) |
| | 3000 | 4500 / 9.4 | N.R. | N.R. | N.R. | N.R. | 0.05±0.04 kg $N_2O$-N ha$^{-2}$y$^{-1}$ November 2010-August 2012[1] | | |
| French Guiana | 49 | 2200 / 26 | 99.4 July-September 1994 | N.R. | N.R. | N.R. | N.R. | N.R. | (Janssens et al., 1998) |
| French Guiana | 30 | 2771.2±628.8 / 27.3±0.5 | N.R. | N.R. | N.R. | N.R. | 15.83±2.1 April 2010-April 2011 | 1.32 [5] | (Petitjean et al., 2015) |
| French Guiana | 147-194 | 2990-3041 / 25.7 | *Nouragues forest* Dry season: 92.6±34.3 (top hill), 89.9±37.8 (middle slope), 131.0±64.2 (bottom slope) October 2015 Wet season: 159±36.5 (top hill), 191.7±66.5 (middle slope), 188.8±50.2 (bottom slope) May 2016 | N.R. | *Nouragues forest* Dry season: -64.0±69.7 (top hill), 6.6±237.9 (middle slope), -49.9±50.1 (bottom slope) October 2015) Wet season: -19.9±70.3 (top hill), 43.2±274.0 (middle slope), 9.4±64.9 (bottom slope) May 2016 | N.R. | *Nouragues forest* Dry season: -20.4±15.0 (top hill), -20.1±17.4 (middle slope), -32.5±21.6 (bottom slope) October 2015 Wet season: -30.7±30.9 (top hill), -31.9±15.5 (middle slope), -55.4±47.6 (bottom slope) May 2016 | N.R. | (Courtois et al., 2018) |

| | | | | | | | | | |
|---|---|---|---|---|---|---|---|---|---|
| | | | *Paracou forest* Dry season: 165.2±50.2 (top hill), 131.7±34.3 (middle slope), 189.6±96.8 (bottom slope) October 2015 Wet season: 161.6±62.9 (top hill), 138.3±88.7 (middle slope), 94.8±57.4 (bottom slope) May 2016 | N.R. | *Paracou forest* Dry season: -44.0±139.7 (top hill), -19.9±79.5 (middle slope), 6.6±103.7 (bottom slope) October 2015 Wet season: 3.7±40.1 (top hill), -1.9±41.2 (middle slope), 23.9±34.6 (bottom slope) May 2016 | N.R. | *Paracou forest* Dry season: -31.2±27.5 (top hill), -41.4±54.5 (middle slope), -35.7±21.8 (bottom slope) October 2015 Wet season: -49.7±49.8 (top hill), -19.3±45.1 (middle slope), -18.0±70.9 (bottom slope) May 2016 | N.R. | |
| **French Guiana** | 40. | 2929[4] 27 | Dry season: 99.6±7.9 Wet season: 111.3±5 May 2011-November 2014 | N.R. | Dry season: -12.9±10.8 Wet season: -12.1±9.2 May 2011-November 2014 | N.R. | Dry season: 8.3±2.1 Wet season: 12.9±1.7 May 2011-November 2014 | N.R. | (Petitjean et al., 2019) |
| **Peru** | 180 | 2200 26 | N.R. | N.R. | -29.0 to -32.1 October 1997-October 1999 | -2.6 | 8.1 to 18.8 October 1997-October 1999 | 0.80 | (Palm et al., 2002) |
| | 200 | 2700 26.4 | Dry season: 0.19±0.01[6] 2007 | N.R | N.R | N.R | N.R | N.R | |
| | 1000 | 3100 21.3 | Dry season: 0.18±0.007[6] 2007 | N.R | N.R | N.R | N.R | N.R | |
| **Peru** | 1500 | 2600 18.3 | Dry season: 1.17±0.007[6] 2007 | N.R | N.R | N.R | N.R | N.R | (Zimmermann et al., 2009) |
| | 3030 | 1700 12.5 | Dry season: 0.18±0.008[6] 2007 | N.R | N.R | N.R | N.R | N.R | |
| **Peru** | 600 - 1200 | 5318 | N.R. | N.R. | Dry season: -13.3±4.6 | -0.14±0.12 | Dry season: 40.0±19.6 | 0.54±0.32 | |

| | | | | | | | | | |
|---|---|---|---|---|---|---|---|---|---|
| | | 23.4 | | | Wet season: 21.3±17.1 July 2011 – December 2011 | | Wet season: -6.3±17.9 July 2011 – December 2011 | | |
| | 1200 - 2200 | 2631 18.8 | N.R. | N.R. | Dry season: -35.0±2.9 Wet season: -20.4±5.4 December 2010 - December 2011 | -0.69±0.09 | Dry season: 40.8±9.6 Wet season: 6.7±5.4 December 2010 - December 2011 | 0.50±0.13 | (Teh et al., 2014) |
| | 2200 - 3200 | 1706 12.5 | N.R. | N.R. | Dry season: -50.8±1.7 Wet season: -22.5±4.6 December 2010 - December 2011 | 0.80±0.08 | Dry season: 7.9±7.1 Wet season: 0.8±9.2 December 2010 - December 2011 | 0.12±0.13 | |
| **Peru** | 1070 - 1088 | 5300 23.4 | Dry season: 216.7±12.5 Wet season: 212.5±12.5 July 2011-June 2013 | N.R. | Dry season: -8.3±4.2 Wet season: -4.2±4.2 July 2011-June 2013 | N.R. | N.R. | N.R. | |
| | 1532 - 1769 | 2600 18.8 | Dry season: 179.2±12.5 Wet season: 170.8±12.5 February 2011-June 2013 | N.R. | Dry season: -45.8±4.2 Wet season: -1.34±4.2 February 2011-June 2013 | N.R. | N.R. | N.R. | (Jones et al., 2016) |
| | 2811 - 2962 | 1700 12.5 | Dry season: 210.8±12.5 Wet season: 166.7±12.5 February 2011-June 2013 | N.R. | Dry season: -66.7±4.2 Wet season: -45.8±4.2 February 2011-June 2013 | N.R. | N.R. | N.R. | |

[1]Measured mean annual fluxes.

[2]Soil-atmosphere fluxes of $CO_2$ and $CH_4$ were measured weekly during June-August 2000, but average values are not reported.

[3]Measured $CO_2$ fluxes from the soil surface.

[4]Average of the annual rainfall measured at the experimental site during 4 years.

[5]Cumulative $N_2O$ fluxes from the 11/05/2010 to the 09/05/2011

[6]Values for undistrubed "native" soil. This study evaluated the climate dependence of heterotrophic soil respiration from a soil-translocation experiment. As a matter of comparison with the others studies, only control cores are depicted, i.e. soil cores re-installed at the same site.

**Table S2.** Characteristics of the study areas Río Silanche (400 m a.s.l.; S_400), Milpe (1100 m a.s.l.; M_1100), El Cedral (2200 m a.s.l.; C_2200) and Peribuela (3010 m a.s.l.; P_3010), including mean annual precipitations (MAP) and mean annual temperatures (MAT) extracted from the Worldclim data set, using average monthly data from 1970-2000 with a spatial resolution of ~1 km$^2$ (Fick and R.J. Hijmans, 2017). Forest classification has been done based on the system used by the country (FAO, 2017; Ministerio del Ambiente, 2015).

| Study area | Forest type | Coordinates | | Altitude (m a.s.l.) | MAP (mm) | MAT (°C) |
|---|---|---|---|---|---|---|
| | | Latitude | Longitude | | | |
| **S_400** | Lowland evergreen forest of Choco | 00°08'45.58'' N | 79°08'34.22'' W | 400 | 3633 | 23.0 |
| **M_1100** | Andean foothill evergreen | 00°02'07.17'' N | 78°51'59.72'' W | 1100 | 2856 | 21.1 |
| **C_2200** | Andean montane evergreen | 00°06'47.87'' N | 78°34'10.88'' W | 2200 | 1464 | 16.8 |
| **P_3010** | Upper montane evergreen | 00°22'27.35'' N | 78°18'0.36'' W | 3010 | 956 | 12.8 |

Note: the coordinates were taken at the center of the plots.

---

## Author Response (AR4)

**Response to the associate editor on the manuscript: "*Ideas and perspectives: patterns of soil CO$_2$, CH$_4$ and N$_2$O fluxes along an altitudinal gradient - a pilot study from an Ecuadorian Neotropical montane forest*"**

Paula Alejandra Lamprea Pineda[1], Marijn Bauters[2, 3], Hans Verbeeck[3], Selene Baez[4], Matti Barthel[5], Samuel Bodé[2] and Pascal Boeckx[2]

We thank the editor for the final assessment of our manuscript and the reviewers for providing constructive comments and suggestions to improve it in the previous stages. The whole process has been successful, and our manuscript improved significantly through it. Therefore, the manuscript has been accepted for publication and no additional corrections are required.